# A conserved mycobacterial nucleomodulin hijacks the host COMPASS complex to reprogram pro-inflammatory transcription and promote intracellular survival

Liu Chen[1,2], Baojie Duan[1], Pingping Chen[1], Qiang Jiang[1], Yifan Wang[1], Lu Lu[1], Yingyu Chen[1,3,4], Changmin Hu[1,4], Lei Zhang[1,3,4]*, Aizhen Guo[1,2,3,4]*

[1]National Key Laboratory of Agricultural Microbiology, College of Veterinary Medicine, Huazhong Agricultural University, Wuhan, China; [2]Hubei Hongshan Laboratory, Huazhong Agricultural University, Wuhan, China; [3]Hubei Jiangxia Laboratory, Wuhan, China; [4]National Professional Laboratory for Animal Tuberculosis, Ministry of Agriculture and Rural Affairs, Huazhong Agricultural University, Wuhan, China

**\*For correspondence:**
zhanglei2023@mail.hzau.edu.cn (LZ);
aizhen@mail.hzau.edu.cn (AG)

**Competing interest:** The authors declare that no competing interests exist.

## eLife Assessment

This **valuable** study provides **convincing** evidence that MgdE, a conserved mycobacterial nucleomodulin, downregulates inflammatory gene transcription by interacting with the histone methyltransferase COMPASS complex and altering histone H3 lysine methylation. This work will interest microbiologists as well as cell and cancer biologists.

**Abstract** Nucleomodulins are a class of effector proteins secreted by bacterial pathogens that translocate into the host cell nucleus to modulate nuclear processes. However, their target proteins and underlying molecular mechanisms remain poorly understood in mycobacteria. Herein, we identified a conserved hypothetical protein Rv1075c, designated MgdE, as a nucleomodulin that enhances mycobacterial intracellular survival. MgdE undergoes nuclear translocation via two nuclear localization signals, KRIR$^{108-111}$ and RLRRPR$^{300-305}$, and interacts with ASH2L and WDR5, two subunits of the host histone methyltransferase COMPASS complex. This interaction suppresses histone H3 lysine 4 (H3K4) methylation-mediated transcription of pro-inflammatory genes, including *IL6* and *IL1B*, thereby promoting mycobacterial survival in both macrophages and mice. Our study provides the first experimental evidence that a bacterial nucleomodulin facilitates intracellular survival by directly targeting the host COMPASS complex. These findings advance our understanding of mycobacterial pathogenesis by revealing a novel mechanism that contributes to its intracellular survival strategy.

## Introduction

Nucleomodulins constitute a distinct class of bacterial proteins capable of translocating into the nuclei of eukaryotic cells, where they strategically modulate nuclear processes to promote pathogen survival and persistence (*Radoshevich and Cossart, 2018*; *Hanford et al., 2021*). These evolutionarily diverse effector proteins, secreted by distinct pathogens, exhibit remarkable adaptability in

infiltrating host nuclei and reprogramming the transcriptional and epigenetic regulatory networks (*Bierne and Cossart, 2012*; *Bierne and Pourpre, 2020*). Upon nuclear localization, nucleomodulins employ diverse molecular mechanisms to subvert host cellular processes, including but not limited to: (i) direct interactions with genomic DNA or nuclear proteins (*Rennoll-Bankert et al., 2015*; *Sun et al., 2016*; *Prokop et al., 2017*), (ii) hijacking post-translational modification systems (*Lebreton et al., 2014*; *Rolando et al., 2023*), (iii) dysregulating signaling molecule activity or localization (*Zhao et al., 2019*; *Chambers and Scheck, 2020*; *Evans et al., 2018*), and (iv) destabilizing nuclear architecture (*Pourpre et al., 2022*; *Fu et al., 2021*). Specific examples include AnkA, a bacterial protein from *Anaplasma phagocytophilum*, which binds to specific A/T-rich sequences in the host cell, alters chromatin architecture, and modulates gene transcription (*Rennoll-Bankert et al., 2015*). In *Listeria monocytogenes*, LntA interacts with the proline-rich domain of BAHD1 to disrupt BAHD1-mediated gene silencing and promote histone H3 acetylation (*Lebreton et al., 2014*). Another example is OspF from *Shigella flexneri*, which exploits its phosphothreonine lyase activity to irreversibly dephosphorylate MAPKs, thereby preventing the transcriptional activation of NF-κB-regulated genes (*Chambers and Scheck, 2020*). Unlike these effectors, *L. monocytogenes* InlP takes a different approach by hijacking the splicing factor RBM5. This interaction not only suppresses apoptosis but also reorganizes nuclear speckles through SC35 redistribution and RBM5-dependent granule formation (*Pourpre et al., 2022*). These examples illustrate the diverse and sophisticated mechanisms used by nucleomodulins to reprogram host nuclear functions, ultimately manipulating the intracellular environment to their advantage.

The evolutionary refinement of nucleomodulins is exemplified by their ability to exploit chromatin-based regulatory systems, with histone methylation emerging as a pivotal axis in host-pathogen interactions (*Bierne and Pourpre, 2020*; *Khan and Khan, 2021*). As a central epigenetic modification, histone methylation dynamically controls chromatin accessibility and transcriptional outcomes (*Yu and Lesch, 2024*; *Wang and Helin, 2025*). By precisely targeting specific histone residues, nucleomodulins reprogram chromatin states to either suppress host immune responses or promote cellular pathways conducive to pathogen survival (*Lebreton et al., 2014*; *Schator et al., 2023*; *Rolando et al., 2023*; *Denzer et al., 2020*). A paradigm of this molecular strategy is demonstrated by the *Chlamydia trachomatis* nuclear effector NUE, which utilizes its conserved SET domain to catalyze the methylation of histones H2B, H3, and H4, establishing a repressive chromatin state that downregulates antimicrobial gene expression (*Fol et al., 2020*). Similarly, RomA uses its ankyrin and SET domains to bind and methylate histone H3, thereby altering the host chromatin to enhance *Legionella* intracellular survival (*Schator et al., 2023*; *Rolando et al., 2013*). *Legionella* LegAS4 catalyzes H3K4 methylation to promote an open chromatin state, enhancing ribosomal RNA transcription and facilitating bacterial intracellular replication (*Denzer et al., 2020*). These cases emphasize how pathogens exploit histone methylation as a molecular lever to hijack the host epigenetic machinery.

*Mycobacterium tuberculosis* (Mtb), the causative agent of tuberculosis, is a highly successful pathogen because of its ability to survive within macrophages, evade the immune system, and persist in the host, ultimately leading to a chronic disease state (*Chandra et al., 2022*; *Chai et al., 2020*). Central to its pathogenic strategy is the secretion of a diverse array of effector proteins (*Chai et al., 2022*; *Bates et al., 2024*; *Qiang et al., 2023*), including nucleomodulins that subvert host nuclear processes. The tyrosine phosphatase PtpA exemplifies this strategy by directly binding to the promoter regions of immune-related genes, including *GADD45A*, to suppress innate immune responses (*Wang et al., 2017*). The methyltransferase Rv1988 mediates an unconventional histone modification by methylating histone H3 at arginine 42 (H3R42me), thereby silencing antimicrobial defense mechanisms (*Yaseen et al., 2015*). Similarly, the acetyltransferase Rv3423.1 modulates chromatin accessibility through acetylation of histone H3 at lysine 9 and 14 (H3K9/K14ac), attenuating pro-inflammatory signaling pathways (*Jose et al., 2016*). Rv2966c represents a unique bacterial DNA methyltransferase that establishes non-canonical CpG methylation patterns to epigenetically silence host immune-related gene expression (*Sharma et al., 2015*). Despite these molecular insights, the comprehensive landscape of the nuclear effector systems of Mtb, particularly the mechanistic details of nucleomodulin interactions with host chromatin-modifying complexes and their spatiotemporal regulation during infection, remains poorly defined. This critical knowledge gap impedes our understanding of how Mtb coordinates system-level transcriptional rewiring in the host cellular environment.

In this study, we utilized bioinformatics analysis combined with fluorescence imaging to rapidly screen for potential nucleomodulins in pathogenic mycobacterial species. Using this approach, the

hypothetical protein Rv1075c (designated as MgdE) was identified and subsequently validated. MgdE undergoes nuclear translocation mediated by two nuclear localization signals, KRIR[108-111] and RLRR-PR[300-305], and interacts with the histone methyltransferase COMPASS complex subunits ASH2L and WDR5. This interaction suppresses H3K4me3-mediated inflammatory gene expression, thus dampening host immune responses and enhancing bacterial survival in macrophages and murine infection models. Our findings identify MgdE as a nucleomodulin that targets the COMPASS complex, revealing a novel epigenetic strategy exploited by mycobacteria for intracellular survival.

## Results

### Functional screening of potential nucleomodulins in pathogenic mycobacterial species

To identify conserved nucleomodulins in pathogenic mycobacteria, we first conducted a comparative pan-genome analysis across four relevant species, including *M. tuberculosis* H37Rv and H37Ra, *M. bovis* BCG, *M. marinum*, and *M. avium*, to categorize the core genes. Subsequently, a dual-threshold screening approach was employed to predict the classical and non-canonical secreted proteins encoded by these shared genes using SignalP 5.0 and SecretomeP 2.0, with thresholds set at a D-score ≥0.5 (*Hammond et al., 2018*) and a neural network (NN) score ≥0.9 (*Bendtsen et al., 2005*; *Supplementary file 1*). This analysis yielded 135 high-confidence secreted protein candidates. Further analysis of these candidate nuclear localization signal (NLS) motifs using cNLS Mapper (NLS score ≥2) identified 56 proteins containing at least one putative NLS (*Supplementary file 2*). Following systematic classification based on secretion machinery dependencies, the candidate proteins were categorized into four distinct pathways: 26 Sec/SPI (signal peptide-driven), 7 Tat/SPI (twin-arginine translocase pathway), 18 Sec/SPII (lipoprotein-targeted), and five unclassified candidates lacking canonical secretion motifs (*Figure 1A and B*, *Figure 1—figure supplement 1A and B*). Functional validation of nuclear trafficking was conducted through the heterologous expression of enhanced green fluorescent protein (EGFP)-tagged constructs in HEK293T cells, followed by quantitative profiling of the subcellular distribution using laser scanning confocal microscopy (LSCM). Among them, six Tat/SPI-associated proteins, including Rv0519c, Rv0846c, Rv0945, Rv1075c/MgdE, Rv1291c, and Rv2577 (*Supplementary file 3*) exhibited nuclear localization, and MgdE exhibited distinct nuclear aggregation (*Figure 1C*). Nucleomodulins are effector proteins secreted by pathogens that translocate into the host cell nucleus to modulate host functions. To examine whether MgdE is secreted by mycobacteria, recombinant *M. bovis* BCG strains expressing C-terminally Flag-tagged MgdE were constructed. Immunoblotting showed that MgdE and the secreted antigen Ag85B were present in both cell pellets and culture supernatants, while the cytoplasmic protein GlpX, as the negative control, was only detected in the cell pellets (*Figure 2—figure supplement 1A*), indicating that MgdE is indeed secreted by *M. bovis* BCG. Collectively, these results suggest that MgdE might be a potential nucleomodulin in the pathogenic mycobacteria.

### MgdE translocates into the host nucleus through its dual NLS

MgdE is a putative GDSL-like lipase characterized by conserved esterase sequence motifs, with the GDSx motif containing a nucleophilic Ser residue as part of block I (equivalent to the classical G-x-S-x-G motif of lipases/esterases), and the consensus amino acids Gly, Asn, and His located in blocks II, III, and V, respectively (*Yang et al., 2019*). Phylogenetic analysis across the mycobacterial species revealed strict evolutionary conservation of MgdE, with most strains encoding two predicted NLS: KRIR[108-111] (NLS1) and RLRRPR[300-305] (NLS2) (*Figure 2A and B*). To further investigate the nuclear translocation capability of MgdE, HEK293T cells were transfected with MgdE-EGFP, and its subcellular localization was examined at multiple time points post-transfection. Confocal microscopy revealed a time-dependent increase of MgdE-EGFP in the nucleus by 48 hr post-transfection (*Figure 2—figure supplement 1B and C*). To quantitatively assess this process, MgdE expression and nuclear enrichment were analyzed by cell fractionation and immunoblotting. These results demonstrated that MgdE protein levels increased over time and progressively accumulated in the nucleus, consistent with the confocal imaging observations (*Figure 2—figure supplement 1D–E*). Taken together, these experiments indicate that MgdE is an evolutionarily conserved mycobacterial nucleomodulin with nuclear translocation capacity.

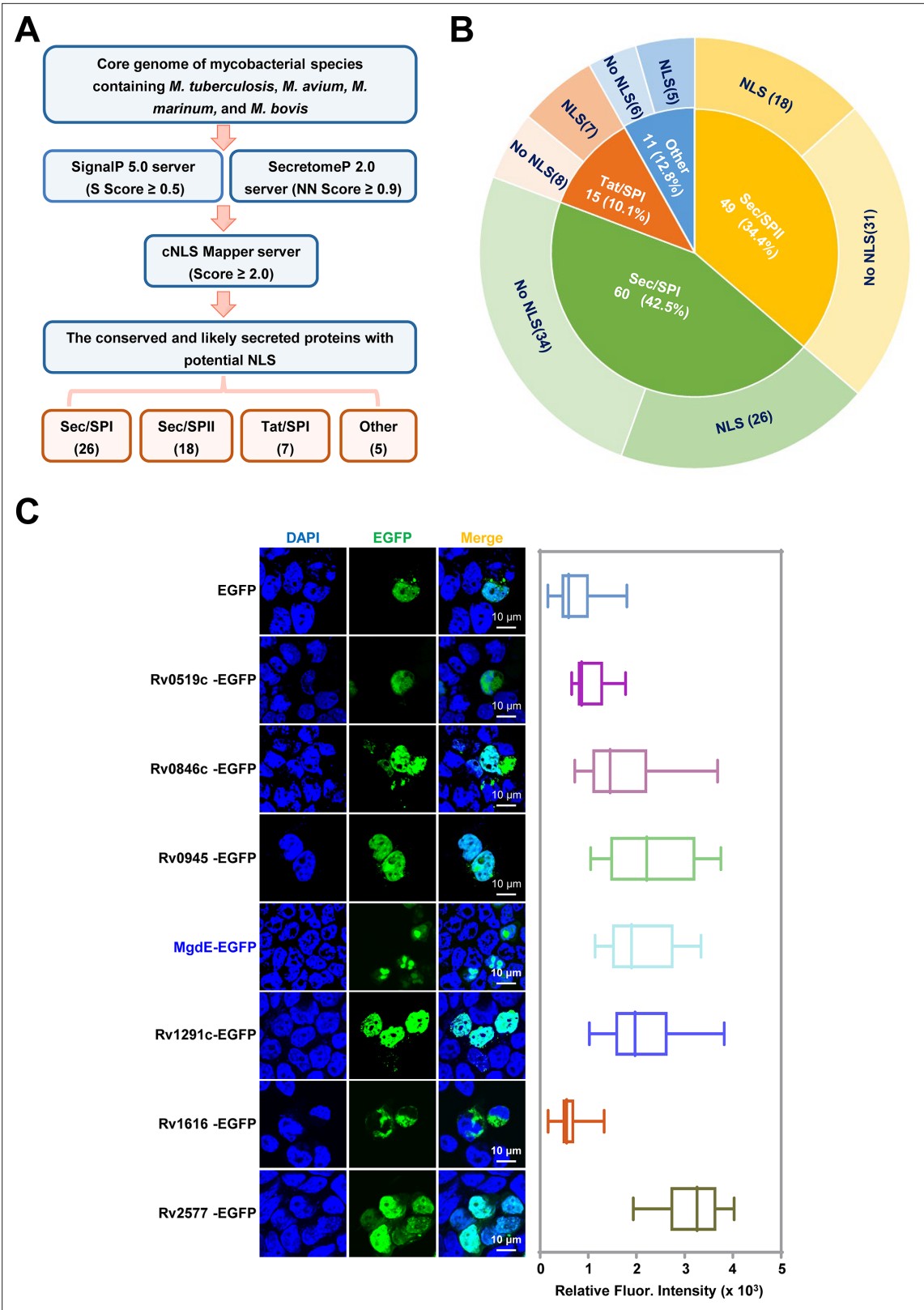

**Figure 1.** Identification of conserved nucleomodulins in mycobacteria through functional screening. (**A**) Schematic representation of the bioinformatic pipeline for identifying conserved nucleomodulins in Mycobacterium species. Genomic sequences from *M. tuberculosis* H37Rv, *M. tuberculosis* H37Ra, *M. avium*, *M. marinum*, and *M. bovis* BCG were analyzed. Signal peptides were predicted using SignalP 5.0 (D-score ≥0.5), non-classical secretion signals were identified using SecretomeP 2.0 (NN score ≥0.9), and nuclear localization signals (NLS) were predicted using cNLS Mapper (score ≥2.0).

*Figure 1 continued on next page*

Figure 1 continued

(**B**) Classification of conserved cellular nucleomodulins. Secreted proteins were categorized based on the presence or absence of predicted NLS motifs. (**C**) Subcellular localization of enhanced green fluorescent protein (EGFP)-tagged candidate nucleomodulins. (left) Confocal microscopy images of seven Tat/SPI proteins fused to EGFP (green), nuclei were stained with DAPI (blue). (right) Quantification of nuclear EGFP fluorescence intensity. Scale bar, 10 μm. Data are presented as mean ± SD (n=25 cells/group).

The online version of this article includes the following figure supplement(s) for figure 1:

**Figure supplement 1.** Comparative analysis of classical and non-classical secreted proteins in mycobacterial species.

To dissect the functional contributions of the predicted NLS motifs, EGFP-tagged NLS deletion mutants, including MgdE$^{\Delta NLS1}$, MgdE$^{\Delta NLS2}$, and MgdE$^{\Delta NLS1-2}$ were constructed and transfected into HEK293T cells. Fluorescence intensity profiling revealed that individual deletion of either NLS1 or NLS2 partially reduced the nuclear targeting efficiency, whereas simultaneous deletion of both the NLS motifs (MgdE$^{\Delta NLS1-2}$) completely abrogated its nuclear localization (*Figure 2C and D*). These observations were further confirmed by immunoblot analysis of nuclear fractions, which showed that at 36 hr post-transfection, MgdE-EGFP and its mutants were consistently detected in the cytoplasmic compartment, and nuclear accumulation of MgdE$^{\Delta NLS1-2}$ was nearly undetectable compared to that of MgdE (*Figure 2E*, *Figure 2—figure supplement 1F*). To determine whether MgdE translocates into the host nucleus during mycobacterial infection, recombinant BCG strains expressing Flag-tagged MgdE mutants were constructed, including single and double NLS deletion mutants (BCG/MgdE$^{\Delta NLS1}$-Flag, BCG/MgdE$^{\Delta NLS2}$-Flag, and BCG/MgdE$^{\Delta NLS1-2}$-Flag). THP-1 macrophages were infected with the recombinant strains, followed by subcellular fractionation. Nuclear-cytoplasmic fractionation experiments showed that WT MgdE and the NLS single mutants could be detected both in the cytoplasm and in the nucleus by immunoblotting, while the double mutant MgdE$^{\Delta NLS1-2}$ was detectable only in the cytoplasm (*Figure 2F*). These results indicate that MgdE translocates into the host nucleus via a dual NLS mechanism, with both NLS1 and NLS2 required for efficient nuclear import.

## Nuclear localization of MgdE enhances mycobacterial intracellular survival within macrophages

Given that MgdE exhibits nuclear translocation ability, we sought to determine whether this ability plays a role in bacterial survival during infection. We generated various recombinant *M. bovis* BCG strains, including wild-type BCG (WT), a *mgdE* deletion mutant (ΔMgdE) (*Figure 3—figure supplement 1A and B*), ΔMgdE complemented with WT *mgdE* (Comp-MgdE), and ΔMgdE complemented with NLS-deleted mutant (Comp-MgdE$^{\Delta NLS1-2}$). All strains exhibited similar growth in 7H9 liquid media supplemented with OADC (*Figure 3—figure supplement 1C*). To assess intracellular survival, we infected THP-1 macrophages with each strain and monitored bacterial burden over time. As shown in *Figure 3A*, the ΔMgdE mutant strain exhibited significantly reduced survival compared to the WT strain, with accelerated bacterial clearance over time. This defect was rescued in the Comp-MgdE strain, with intracellular survival levels restored to that in the WT strain. Notably, the Comp-MgdE$^{\Delta NLS1-2}$ showed bacterial clearance equivalent to that in the ΔMgdE strains. These results were consistent with the infection experiments using RAW264.7 macrophages (*Figure 3B*). Furthermore, THP-1 macrophages infected with the ΔMgdE mutant strain exhibited significant upregulation of inflammatory cytokines, including *IL1B*, *IL6*, *IL10,* and the colony-stimulating factors *CSF1-CSF3* compared to the macrophages infected with the WT strain (*Figure 3C–H*). Collectively, these results demonstrated that (i) MgdE enhances mycobacterial intracellular survival and suppresses host inflammatory responses during infection, and (ii) both NLS1 and NLS2 motifs are essential for the nuclear trafficking-dependent function.

## MgdE physically interacts with the host COMPASS complex subunits ASH2L and WDR5

To identify the potential host target proteins of MgdE, we employed AlphaFold to predict the interactions between MgdE and the majority of the nuclear proteins in the host. The prediction results showed that MgdE binds to WDR5, a core component of the histone methyltransferase COMPASS complex, with high confidence (pLDDT = 0.77) (*Figure 4—figure supplement 1A and B*). To experimentally validate this prediction, we performed yeast two-hybrid (Y2H) assays. The yeast cells co-expressing

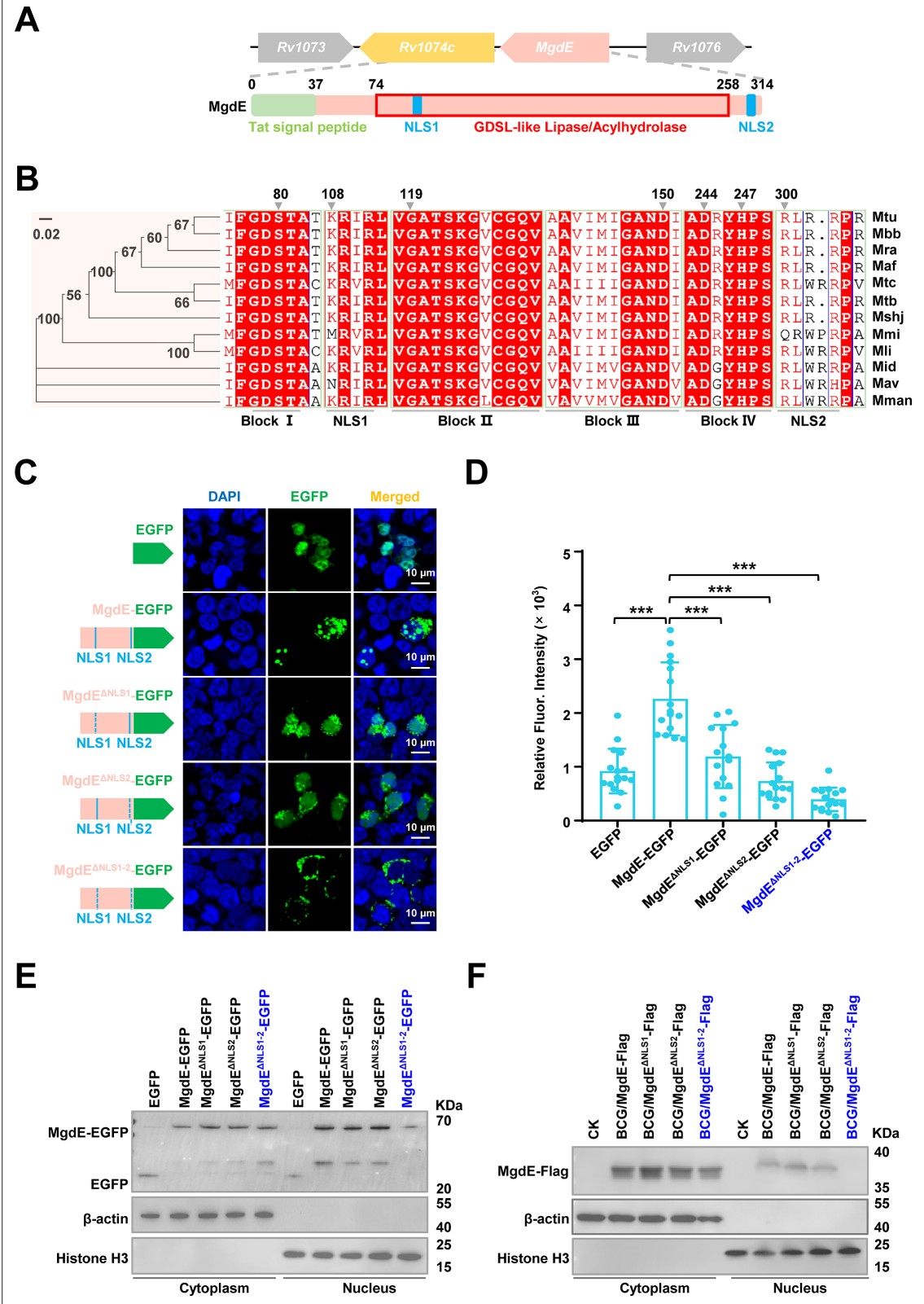

**Figure 2.** Hypothetical protein MgdE enters the host nucleus via dual nuclear localization signal (NLS). (**A**) Domain architecture of MgdE. Schematic representation of MgdE with the annotated functional domains, including a Tat signal peptide (1–37 aa, twin-arginine translocation motif), a GDSL-like lipase/acylhydrolase catalytic domain (74–258 aa), and two nuclear localization signals (NLS1: 108–111 aa, NLS2: 300–305 aa). (**B**) Phylogenetic and structural conservation of MgdE. (left) Neighbor-joining phylogenetic tree of MgdE homologs across Mycobacterium species (1000 bootstrap replicates,

*Figure 2 continued on next page*

*Figure 2 continued*

values ≥50% shown). (right) Clustal Omega sequence alignment highlighting conserved residues (≥90% identity, red). (**C**) Subcellular localization of EGFP-tagged wild-type MgdE and its NLS-deletion mutants (MgdE$^{\Delta NLS1}$, MgdE$^{\Delta NLS2}$, and MgdE$^{\Delta NLS1-2}$). (left) Schematic representation of EGFP-tagged constructs. (right) Representative confocal microscopy images of HEK293T cells transfected with the indicated constructs for 36 hr. Enhanced green fluorescent protein (EGFP) fluorescence (green) and nuclear staining with DAPI (blue) were visualized using FLUOVIEW software. Scale bar, 10 μm. Images were acquired using a 100x oil immersion objective (NA = 1.4). (**D**) Quantification of nuclear EGFP intensity in cells expressing wild-type or mutant MgdE constructs. Data are presented as mean ± SD (n=15 cells). Statistical significance was assessed by two-tailed unpaired Student's *t*-tests (***$p<0.001$). (**E**) Western blot analysis of nuclear and cytoplasmic fractions from HEK293T cells transfected with wild-type MgdE and its NLS-deletion mutants. The empty EGFP vector was used as a negative control. EGFP and MgdE-EGFP fusion proteins were detected using an anti-GFP antibody. Histone H3 and β-actin served as nuclear and cytoplasmic markers, respectively. (**F**) Nuclear localization of MgdE during infection. THP-1 macrophages were infected with recombinant *M. bovis* BCG strains expressing Flag-tagged wild-type MgdE or NLS-deletion mutants (MgdE$^{\Delta NLS1}$, MgdE$^{\Delta NLS2}$, and MgdE$^{\Delta NLS1-2}$) for 24 hr. Subcellular fractionation was performed, and cytoplasmic and nuclear fractions were analyzed by immunoblotting using anti-Flag antibodies. Histone H3 and β-actin were used as nuclear and cytoplasmic markers, respectively.

The online version of this article includes the following source data and figure supplement(s) for figure 2:

**Source data 1.** Original files for western blot analysis displayed in panel E.

**Source data 2.** Original western blots for panel E, indicating the relevant bands.

**Source data 3.** Original files for western blot analysis displayed in panel F.

**Source data 4.** Original western blots for panel F, indicating the relevant bands.

**Figure supplement 1.** Subnuclear localization of MgdE-enhanced green fluorescent protein (EGFP).

**Figure supplement 1—source data 1.** Original files for western blot analysis displayed in panel A.

**Figure supplement 1—source data 2.** Original western blots for panel A, indicating the relevant bands.

**Figure supplement 1—source data 3.** Original files for western blot analysis displayed in panel E.

**Figure supplement 1—source data 4.** Original western blots for panel E, indicating the relevant bands.

MgdE with either ASH2L or WDR5 exhibited robust growth on selective SD/-Trp-Leu-His-Ade medium, whereas the cells co-expressing MgdE with DPY30 or RbBP5 showed no detectable growth, indicating specific interactions between MgdE and ASH2L/WDR5 (*Figure 4A*). Furthermore, these interactions were validated by co-immunoprecipitation (Co-IP) assays. HEK293T cells were co-transfected with Flag-tagged MgdE and HA-tagged ASH2L, WDR5, or RbBP5. The results revealed that HA-tagged ASH2L and WDR5, but not RbBP5, co-precipitated with Flag-tagged MgdE (*Figure 4B*). Conversely, Flag-tagged MgdE precipitated HA-tagged ASH2L and WDR5, but not RbBP5 (*Figure 4C*). Taken together, these results demonstrate that MgdE specifically interacts with the COMPASS complex subunits ASH2L and WDR5.

## The conserved residues D224 and H247 are critical for mediating the binding of MgdE to WDR5

To further confirm the interactions between MgdE and WDR5/ASH2L, the key amino acids of MgdE (S80, D244, and H247) were individually replaced with alanine (A) (*Yang et al., 2019*). The Y2H assays showed that single-residue substitutions of MgdE did not affect its interaction with ASH2L or WDR5. However, the D244AH247A double mutation specifically abrogated its interaction with WDR5 but retained its ASH2L binding ability (*Figure 5A*). These findings were further confirmed by Co-IP assays in HEK293T cells. Flag-tagged MgdE precipitated HA-tagged ASH2L and WDR5 but not RbBP5, with reciprocal pull-downs confirming bidirectional binding specificity. The D244AH247A double mutant maintained its interaction with ASH2L but failed to bind WDR5, suggesting that these interactions are mediated through distinct molecular interfaces (*Figure 5B*, *Figure 4—figure supplement 1C*). To further investigate the functional consequences of these mutations, we analyzed the subcellular localization of the MgdE mutants in transfected HEK293T cells using fluorescence co-localization assays. Wild-type MgdE exhibited distinct nuclear aggregation over time, a phenomenon that was abrogated following transfection with the D244AH247A mutant (*Figure 5C*), suggesting that the double mutation impairs MgdE activity.

Given that MgdE interacts with WDR5, a subunit of the COMPASS complex, which is a key regulator of H3K4me3 methylation (*Deng et al., 2024*), we sought to determine whether this interaction contributes to an epigenetic regulatory mechanism that facilitates bacterial persistence within host cells. HEK293T cells were transfected with EGFP-tagged wild-type MgdE (WT), the catalytic mutant

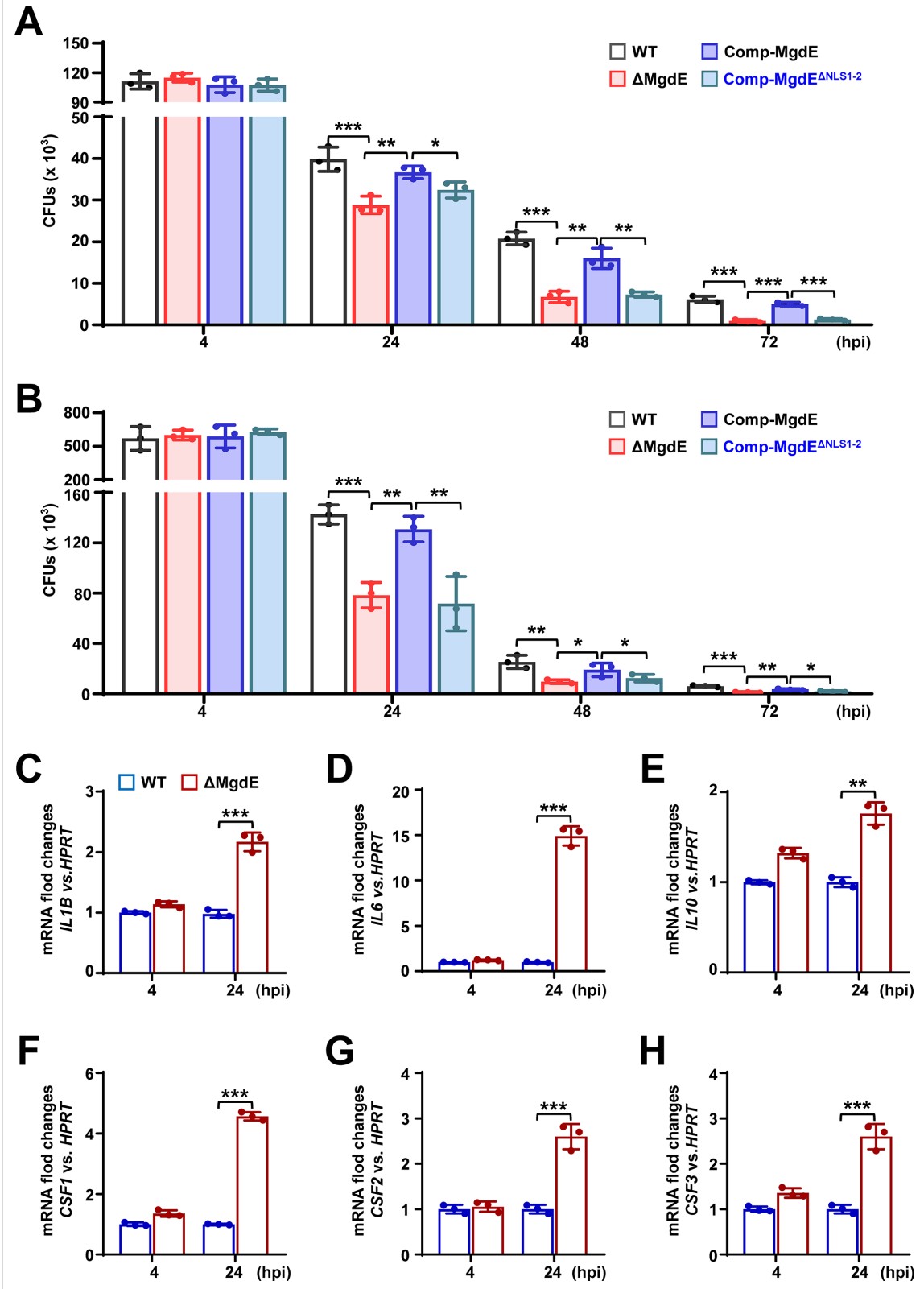

**Figure 3.** Nuclear localization of MgdE facilitates mycobacterial intracellular survival in macrophages. (**A–B**) Intracellular survival of *M. bovis* BCG strains. THP-1 human macrophages (**A**) and RAW264.7 murine macrophages (**B**) were infected (MOI = 10) with wild-type BCG (WT), MgdE deletion mutant (ΔMgdE), complemented strain (Comp-MgdE), or NLS-deficient complement (Comp-MgdE$^{\Delta NLS1-2}$). Bacterial survival was assessed by colony-forming unit (CFU) enumeration at 2, 24, 48, and 72 hr post-infection. (**C–H**) Cytokine expression in the infected THP-1 cells. qRT-PCR analysis of cytokine mRNA

*Figure 3 continued on next page*

*Figure 3 continued*

levels in THP-1 cells infected with WT or ΔMgdE strains for 4-24 hr. Target genes include *IL1B* (**C**), *IL6* (**D**), *IL10* (**E**), *CSF1* (**F**), *CSF2* (**G**), and *CSF3* (**H**). Data represent mean ± SD of three independent biological replicates. Statistical significance was determined using two-way ANOVA or two-tailed unpaired Student's *t*-tests, *$p<0.05$, **$p<0.01$, and ***$p<0.001$.

The online version of this article includes the following source data and figure supplement(s) for figure 3:

**Figure supplement 1.** Deletion of the nuclear localization signal of MgdE does not affect the growth of *M.*

**Figure supplement 1—source data 1.** Original files for western blot analysis displayed in panel B.

**Figure supplement 1—source data 2.** Original western blots for panel B, indicating the relevant bands.

MgdE-D244AH247A, or EGFP protein. Immunoblot analysis of nuclear extracts showed that cells expressing WT MgdE had ~25% lower H3K4me3 levels than EGFP-expressing cells and ~40% lower levels than those expressing the D244AH247A mutant (**Figure 5D**). Taken together, these data indicate that the residues D244 and H247 are essential for the MgdE-WDR5 interaction, which enables MgdE to attenuate H3K4me3 deposition in the host nuclei by subverting COMPASS complex function, a mechanism critical for bacterial survival and immune evasion.

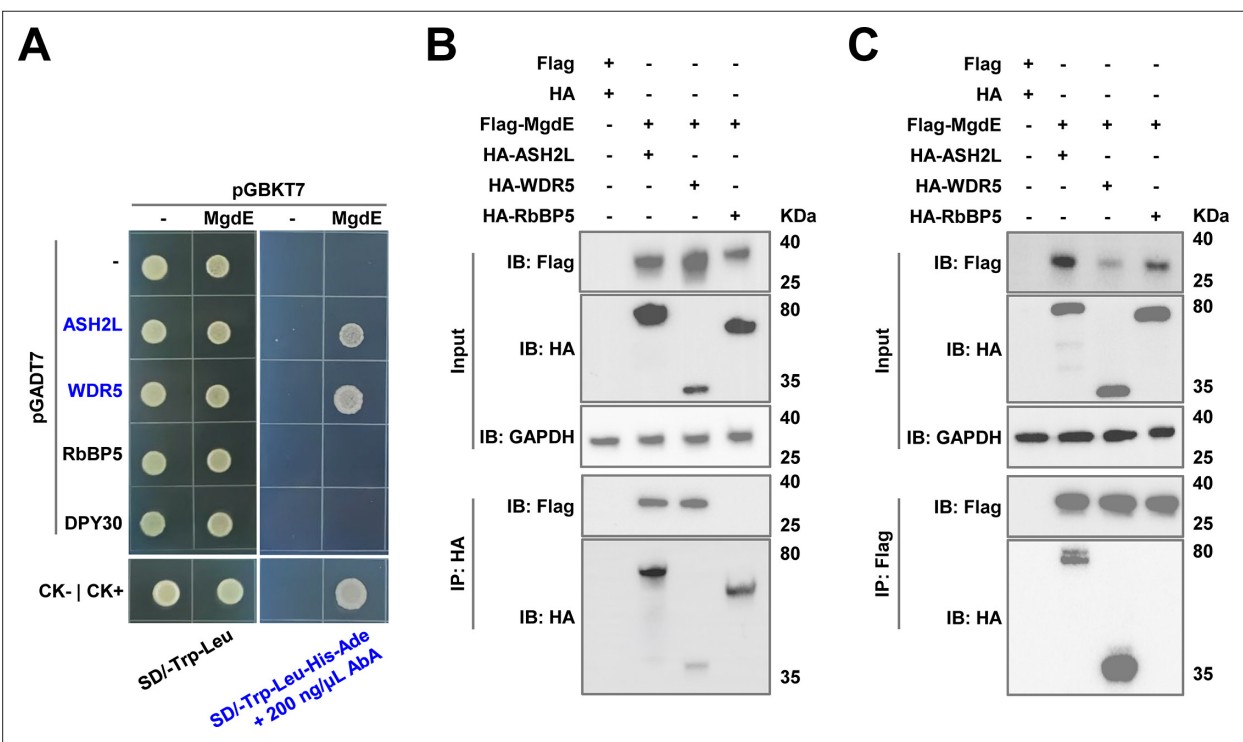

**Figure 4.** MgdE directly interacts with ASH2L and WDR5, core components of the host COMPASS complex. (**A**) Yeast cells were co-transformed with bait (pGBKT7) and prey (pGADT7) plasmids expressing wild-type MgdE and human COMPASS components (ASH2L, WDR5, RbBP5, and DPY30). Growth was monitored on non-selective (-Leu/-Trp, left) and selective (-Leu/-Trp/-Ade/-His+200 ng/μL aureobasidin A, right) media. Controls: CK− (pGBKT7-*lam*+pGADT7 T, negative), CK+ (pGBKT7-*p53*+pGADT7 T, positive). (**B–C**) Cells were co-transfected with Flag-MgdE and HA-tagged ASH2L, WDR5, or RbBP5 (1:1 molar ratio). At 36 hr post-transfection, the lysates were immunoprecipitated using (**B**) anti-HA or (**C**) anti-Flag antibodies, followed by immunoblotting with anti-HA, anti-Flag, and anti-GAPDH (loading control). The input lanes represent 5% of the total lysate.

The online version of this article includes the following source data and figure supplement(s) for figure 4:

**Source data 1.** Original files for western blot analysis displayed in panel B.

**Source data 2.** Original western blots for panel B, indicating the relevant bands.

**Source data 3.** Original files for western blot analysis displayed in panel C.

**Source data 4.** Original western blots for panel C, indicating the relevant bands.

**Figure supplement 1.** MgdE interacts with COMPASS complex subunits.

**Figure supplement 1—source data 1.** Original files for western blot analysis displayed in panel C.

**Figure supplement 1—source data 2.** Original western blots for panel C, indicating the relevant bands.

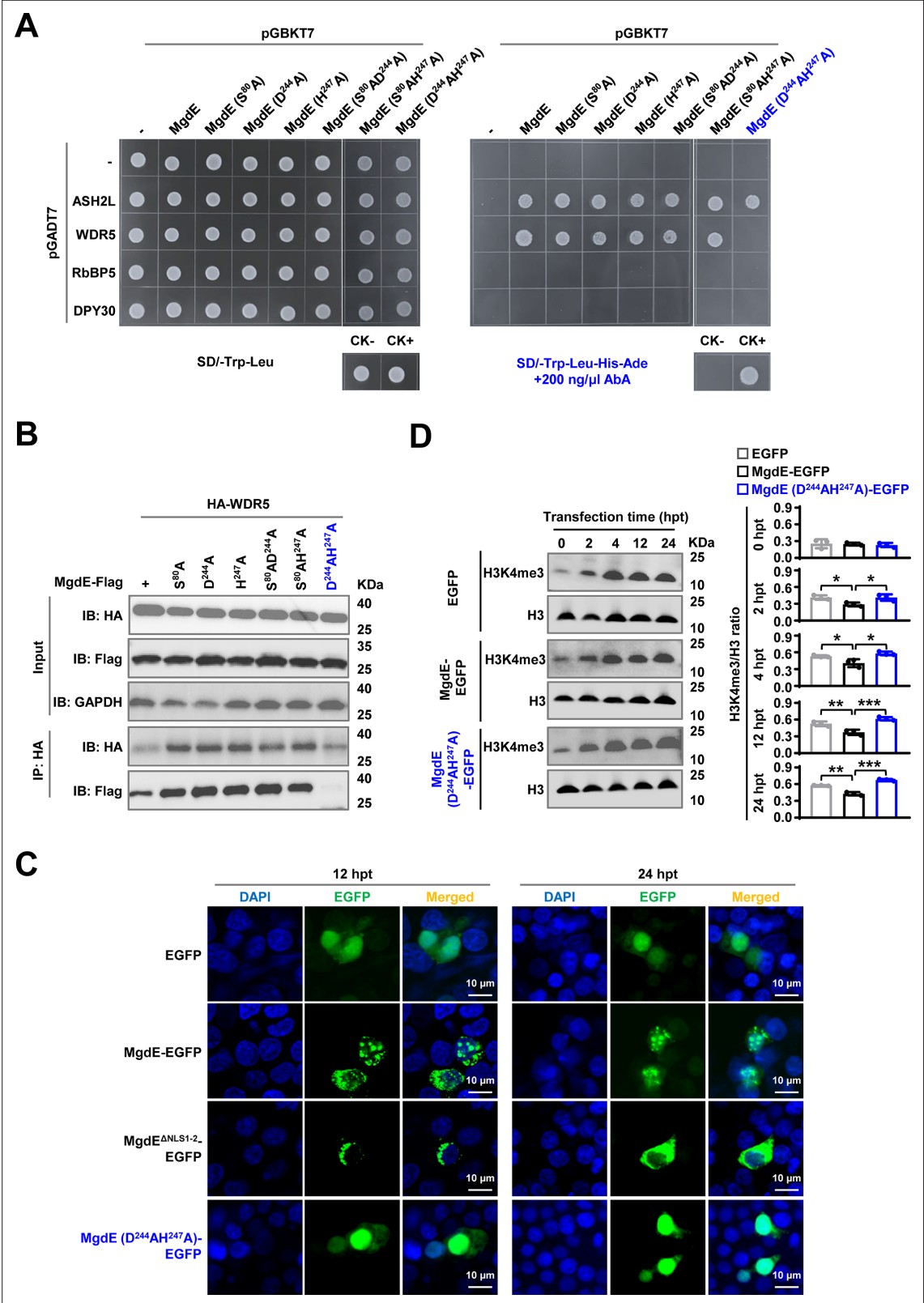

**Figure 5.** The conserved residues D224 and H247 mediate the binding ability of MgdE to WDR5. (**A**) Y2H assay identifying interactions between MgdE mutants and COMPASS complex subunits. Yeast cells were co-transformed with bait (pGBKT7) and prey (pGADT7) plasmids expressing wild-type or mutant MgdE and human COMPASS subunits (ASH2L, WDR5, RbBP5, and DPY30). Growth was assessed on non-selective (-Leu/-Trp, left) and selective (-Leu/-Trp/-Ade/-His+200 ng/µL aureobasidin A, right) media. Controls: CK– (pGBKT7-*lam*+pGADT7 T, negative) and CK+ (pGBKT7-*p53*+pGADT7 T,

*Figure 5 continued on next page*

Figure 5 continued

positive). (**B**) Co-IP analysis of MgdE mutants with WDR5. HEK293T cells were co-transfected with Flag-tagged MgdE mutants and HA-tagged WDR5 (1:1 molar ratio). Complexes were immunoprecipitated using anti-HA antibody and protein A/G beads, followed by immunoblotting with anti-Flag and anti-HA antibodies. (**C**) Nuclear distribution of wild-type and mutants MgdE. Confocal microscopy of HEK293T cells expressing wild-type or D224AH247A MgdE-EGFP at 12 and 24 hr post-transfection (hpt). Nuclear foci were visualized by enhanced green fluorescent protein (EGFP) (green) and DAPI (blue) staining. Scale bar, 10 µm. Images were acquired with a 100x oil immersion objective (NA = 1.4). (**D**) Immunoblot analysis of H3K4me3 levels. HEK293T cells expressing wild-type or D224AH247A mutant MgdE were analyzed for changes in H3K4me3 levels over 0-24 hr post-transfection. Histone H3 was used as a loading control. Data represent mean ± SD of three independent biological replicates. Statistical significance was determined using two-tailed unpaired Student's *t*-tests, *$p<0.05$, **$p<0.01$, ***$p<0.001$.

The online version of this article includes the following source data for figure 5:

**Source data 1.** Original files for western blot analysis displayed in panel B.

**Source data 2.** Original western blots for panel B, indicating the relevant bands.

**Source data 3.** Original files for western blot analysis displayed in panel D.

**Source data 4.** Original western blots for panel D, indicating the relevant bands.

## MgdE suppresses host inflammatory responses probably by inhibition of COMPASS complex-mediated H3K4 methylation

As H3K4me3, a hallmark of transcriptionally active promoters, facilitates transcriptional machinery assembly and RNA polymerase II recruitment (*Wang et al., 2023*), we further investigated the transcriptional consequences of MgdE-mediated H3K4me3 suppression via its interaction with the COMPASS complex. To assess the genome-wide effects of MgdE on host gene expression, we performed RNA-seq profiling of THP-1 macrophages infected with either WT BCG or ΔMgdE strains. Transcriptomic analysis at 24 hr post-infection revealed that 271 genes showed significant differential expression in the cells infected with the ΔMgdE strain compared to those infected with the WT BCG strain, with 222 genes being transcriptionally upregulated (*Figure 6A*). Gene ontology (GO) enrichment analysis indicated that these upregulated genes were primarily associated with biological processes involved in immune responses, with a notable enrichment in 'Positive regulation of cytokine production (GO: 0001819),' a critical pathway for amplifying inflammatory signaling (*Figure 6B*). Furthermore, the Kyoto encyclopedia of genes and genomes (KEGG) pathway enrichment analysis highlighted a significant activation of the 'Cytokine-cytokine receptor interaction (hsa04060)' and 'JAK–STAT signaling pathway (hsa04630)' in THP-1 cells infected with the ΔMgdE strain (*Figure 6C*, *Figure 6—figure supplement 1A*). Heatmap analysis further confirmed the significant upregulation of inflammation-associated transcripts in ΔMgdE-infected cells, including inflammatory cytokines (*IL6*, *IL7*, *IL10*, *IL12B*) and colony-stimulating factors (*CSF2-CSF3*) (*Figure 6D*, *Figure 6—figure supplement 1B*). Protein-protein interaction (PPI) network analysis of the upregulated genes revealed tightly interconnected clusters within inflammatory networks, particularly those orchestrated by the central inflammatory mediator IL-6 (*Figure 6—figure supplement 1C*). To validate this result, we infected macrophages with the *M. bovis* BCG strains, including WT, ΔMgdE, Comp-MgdE, and Comp-MgdE[ΔNLS1-2], under identical conditions for 24 hr and examined the expression of relevant inflammatory factors. Compared to cells infected with the WT and Comp-MgdE strains, those infected with the ΔMgdE and Comp-MgdE[ΔNLS1-2] strains exhibited significantly increased expression levels of inflammatory factors (*IL1B*, *IL6*, *IL10*) (*Figure 6E–G*). Collectively, these results demonstrate that MgdE effectively inhibits the production of inflammatory factors during BCG infection and highlights the critical role of its NLS in this process. Through its interaction with the COMPASS complex, MgdE suppresses H3K4me3 levels, thereby attenuating the transcriptional activation of inflammation-related genes.

## Nuclear localization of MgdE is essential for mycobacterial survival in mice

Our experimental data demonstrated that MgdE enhances bacterial survival in macrophages. To further investigate the in vivo functional significance of MgdE in host-pathogen interplay, we infected C57BL/6 mice with different *M. bovis* BCG strains, including WT, ΔMgdE, Comp-MgdE, and Comp-MgdE[ΔNLS1-2], and comprehensively evaluated bacterial survival, pulmonary inflammation, and cytokine dynamics. Mice infected with ΔMgdE exhibited a significantly reduced lung bacterial burden compared to those infected with WT or Comp-MgdE strains. Moreover, mice infected with Comp-MgdE[ΔNLS1-2]

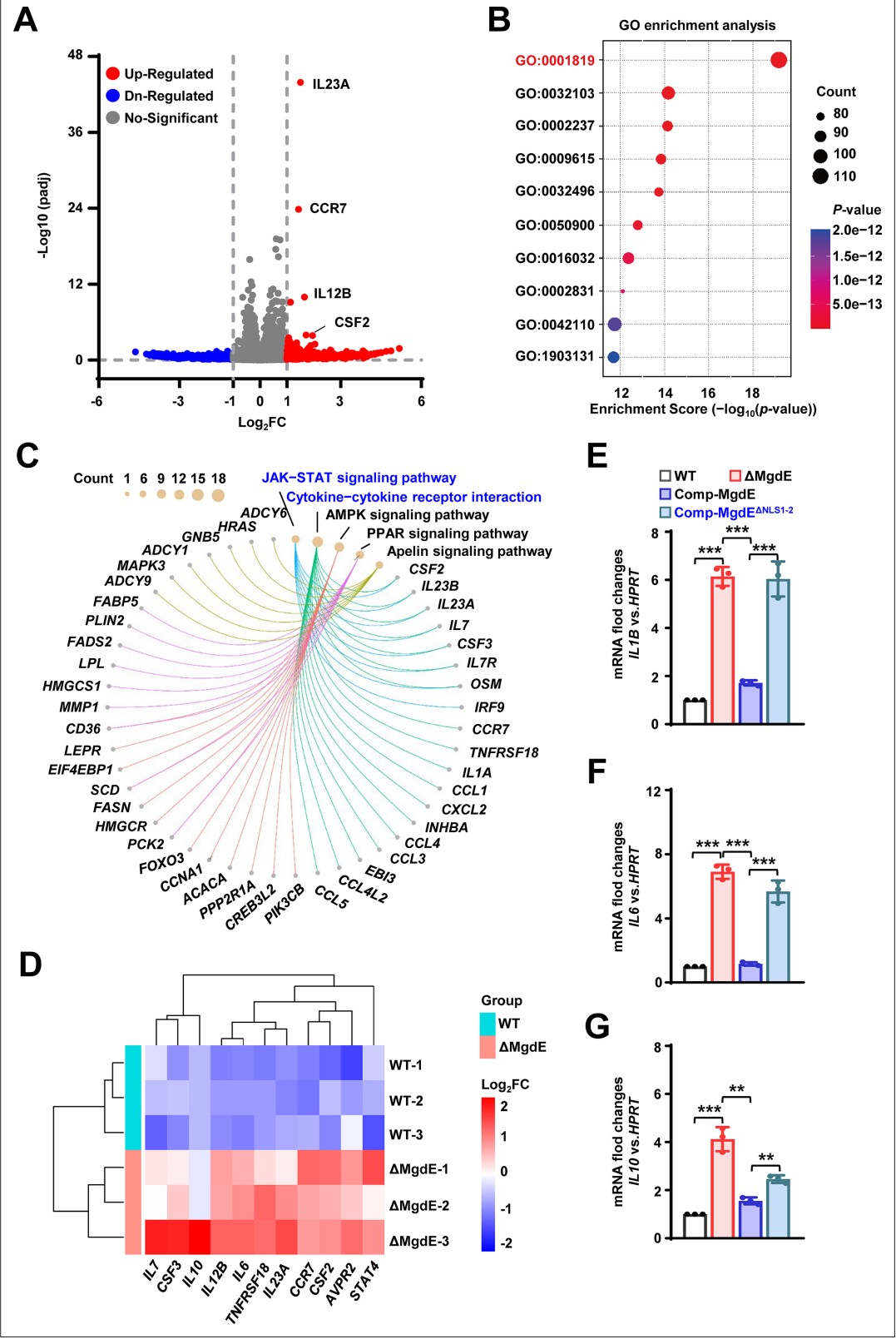

**Figure 6.** MgdE suppresses host inflammatory responses probably by inhibition of COMPASS complex-mediated H3K4 methylation. (**A**) Volcano plot of DEGs. DEGs between MgdE-deleted strain (ΔMgdE) and wild-type BCG (WT) were visualized in a volcano plot. Genes with |log$_2$-fold change|≥1 and $p$<0.05 were considered significant. The $x$-axis represents log$_2$fold change, and the $y$-axis shows -log$_{10}$($p$-value). (**B**) Gene ontology (GO) enrichment

*Figure 6 continued on next page*

*Figure 6 continued*

analysis of DEGs. GO analysis revealed the significant enrichment of immune and inflammatory processes in ΔMgdE-infected macrophages compared to that in WT-infected cells, including positive regulation of response to external stimulus (GO:0032103), response to molecule of bacterial origin (GO:0002237), response to virus (GO:0009615), response to lipopolysaccharide (GO:0032496), leukocyte migration (GO:0050900), viral process (GO:0016032), regulation of response to biotic stimulus (GO:0002831), T cell activation (GO:0042110), and mononuclear cell differentiation (GO:1903131). (**C**) KEGG pathway enrichment analysis of DEGs. Chord diagram of KEGG pathway enrichment analysis showing signaling pathways that are significantly enriched in ΔMgdE. (**D**) Heatmap of the inflammatory gene expression. Heatmap depicting $\log_2$-fold change levels of inflammatory genes involved in the JAK-STAT and cytokine signaling pathways. Upregulated and downregulated genes in ΔMgdE are shown in green and red, respectively. The data were Z-score normalized. (**E–G**) qRT-PCR analysis of cytokine mRNA levels in THP-1 cells infected with WT, ΔMgdE, MgdE-complemented (Comp-MgdE), and nuclear localization signal (NLS)-deleted complement (Comp-MgdE$^{\Delta NLS1-2}$) at 24 hr post-infection. Cytokines analyzed include *IL1B* (**E**), *IL6* (**F**), and *IL10* (**G**). Data represent mean ± SD of three independent biological replicates. Statistical significance was determined using two-tailed unpaired Student's *t*-tests, \*\*$p<0.01$, \*\*\*$p<0.001$.

The online version of this article includes the following figure supplement(s) for figure 6:

**Figure supplement 1.** MgdE suppresses cellular inflammatory responses during *M. bovis* BCG infection.

---

exhibited a significantly lower pulmonary bacterial load than those infected with Comp-MgdE (*Figure 7A*). These findings collectively establish that MgdE is essential for optimal bacterial survival during infection, with the NLS being critical for its full functionality. Furthermore, histopathological assessment of the lung tissues corroborated these observations. Hematoxylin-eosin (H&E) staining revealed more pronounced inflammatory pathology in the lungs of mice infected with WT or Comp-MgdE strain compared to those infected with ΔMgdE or Comp-MgdE$^{\Delta NLS1-2}$ (*Figure 7B*), underscoring the necessity of MgdE and its NLS in sustaining inflammation during infection. To further investigate the systemic immune modulation mediated by MgdE, we quantified splenic bacterial colonization and the inflammatory cytokine profiles. Consistent with the attenuated bacterial survival observed in the lungs, mice infected with ΔMgdE and Comp-MgdE$^{\Delta NLS1-2}$ strains exhibited a significantly reduced splenic bacterial load compared to those infected with WT or Comp-MgdE (*Figure 7—figure supplement 1*). Despite diminished bacterial dissemination, infections with ΔMgdE and Comp-MgdE$^{\Delta NLS1-2}$ strains triggered elevated expression of inflammatory cytokines (*Il6, Il1b*) relative to infection with the WT and Comp-MgdE strains (*Figure 7C and D*). Collectively, these findings indicate that MgdE orchestrates mycobacterial survival within host tissues by suppressing inflammatory responses to evade pathogen clearance, with the NLS playing a pivotal role in mediating these host-pathogen interactions.

## Discussion

### MgdE functions as a critical nucleomodulin that contributes to mycobacterial pathogenesis

Its evolutionary conservation across mycobacterial species, coupled with nuclear translocation via dual N-terminal and C-terminal NLS, supports its classification as a nucleomodulin. Previous investigations have described MgdE as a secreted effector protein (*Penn et al., 2018*), and our experimental results confirm its secretion (*Figure 2—figure supplement 1A*). Furthermore, our findings demonstrate that MgdE's nuclear localization plays a critical role during infection. Disruption of MgdE expression or its NLS motifs significantly attenuated *M. bovis* BCG viability within macrophages (*Figure 3A and B*) and reduced bacterial burden in mice (*Figure 7A*, *Figure 7—figure supplement 1*), supporting the notion that nuclear targeting contributes to bacterial virulence. This phenotype aligns with transcriptomic data showing upregulation of MgdE during *M. tuberculosis* infection of alveolar macrophages (*Pisu et al., 2020*). In parallel, *Yang et al., 2019* characterized MgdE as a lipase involved in intracellular lipid metabolism and showed that its transposon mutant exhibited impaired intracellular growth in THP-1 cells and PBMC-derived macrophages, as well as a moderately reduced bacterial load in C3HeB/FeJ mice at later stages of infection. These findings align with our data and highlight the multifaceted role of MgdE in both metabolic adaptation and host-cell interaction. Together, these findings suggest a functional duality in which MgdE contributes to both metabolic adaptation and host nuclear

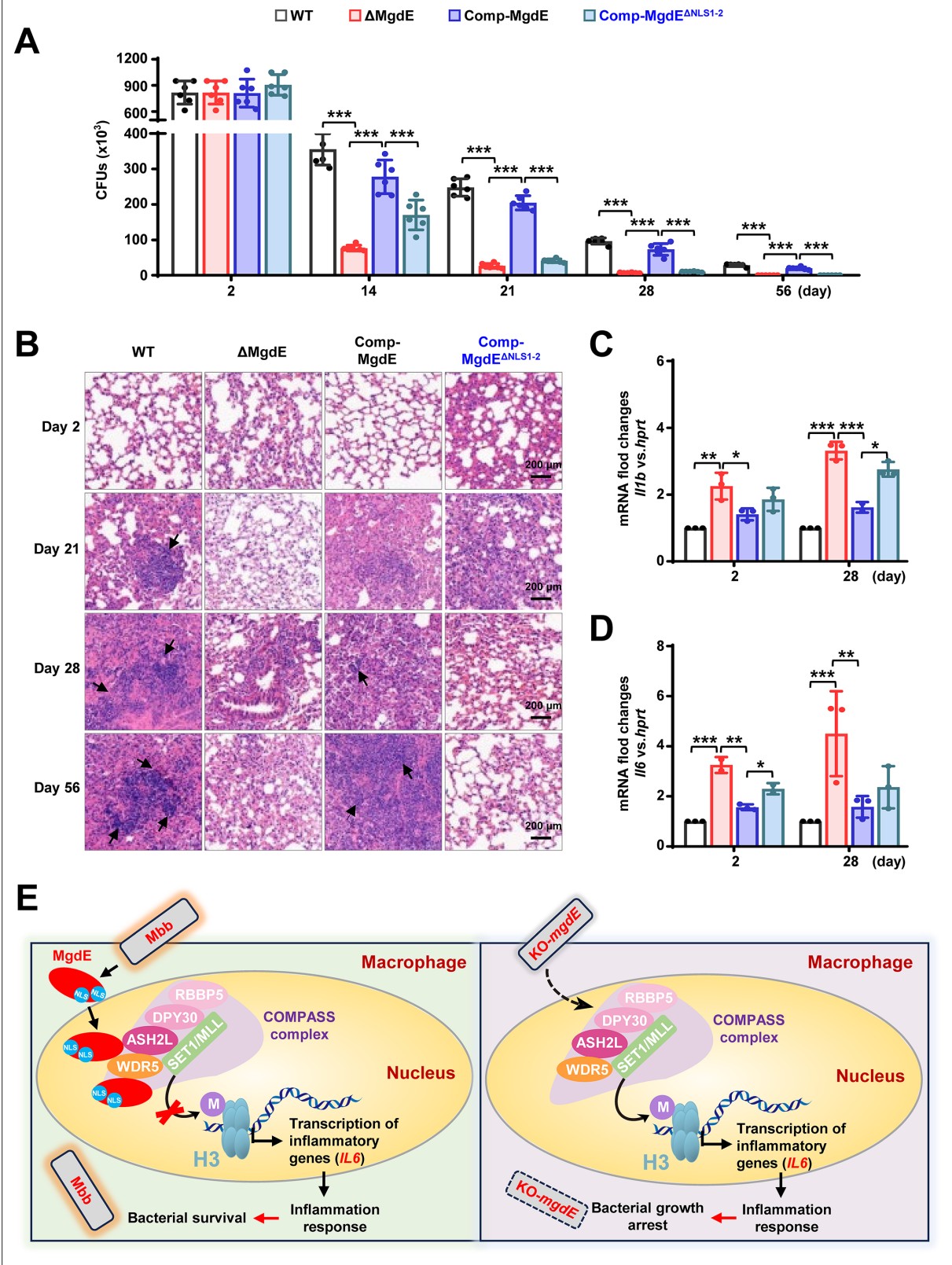

**Figure 7.** Nuclear localization of MgdE is essential for mycobacterial survival in mice. (**A**) Bacterial burden in the lungs of the infected mice. C57BL/6 mice (n=6/group) maintained under specific pathogen-free (SPF) conditions were intratracheally infected with 1.0×10⁷ colony-forming units (CFU) of *M. bovis* BCG strains, including wild-type (WT), MgdE-deleted (ΔMgdE), MgdE-complemented (Comp-MgdE), and nuclear localization signals (NLS)-deleted complement (Comp-MgdE^ΔNLS1-2). Lung bacterial loads were quantified using CFU assays at 0, 14, 21, 28, and 56 days post-infection.

*Figure 7 continued on next page*

*Figure 7 continued*

The data were obtained from a single experiment. (**B**) Hematoxylin and eosin (H&E)-stained lung sections from infected mice (as in **A**) revealed granulomatous inflammation. Scale bars: 200 µm. (**C-D**) Pro-inflammatory cytokine expression in mice spleen. qRT-PCR analysis of cytokine mRNA levels of *Il1b* (**C**) and *Il6* (**D**) in spleen tissues from infected mice (n=6/group) at 2 and 28 days post-infection. Data are presented as mean ± SD from six biologically independent experiments. Statistical significance was determined using two-way ANOVA, *$p<0.05$, **$p<0.01$, ***$p<0.001$. (**E**) Mechanistic model showing how mycobacterial nucleomodulin MgdE hijacks the COMPASS complex to suppress H3K4me3 and promote immune evasion. Upon *M. bovis* BCG infection, the nucleomodulin MgdE is delivered into the host nucleus via its NLS and directly binds to the COMPASS complex subunits, ASH2L or WDR5. This interaction disrupts H3K4 trimethylation (H3K4me3) deposition, leading to the epigenetic suppression of pro-inflammatory cytokine transcription (e.g. *IL6*), thereby facilitating the intracellular survival of the pathogen.

The online version of this article includes the following figure supplement(s) for figure 7:

**Figure supplement 1.** MgdE facilitates bacterial colonization in the spleens of infected mice.

modulation. By coupling enzymatic activity with nuclear targeting, MgdE may enable Mtb to balance metabolic needs and immune evasion.

## MgdE directly subverts the COMPASS complex to reprogram host epigenetics

To our knowledge, MgdE is the first identified bacterial effector that directly targets the COMPASS complex, a central regulator of H3K4 methylation. Nuclear-localized MgdE specifically interacts with the core COMPASS subunits ASH2L and WDR5, disrupting H3K4me3 deposition through structural interference (*Guarnaccia et al., 2021*; *Hsu et al., 2018*; *Xue et al., 2019*). The COMPASS complex facilitates H3K4 methylation through a conserved assembly mechanism involving multiple core subunits. WDR5, a central scaffolding component, interacts with RbBP5 and ASH2L to promote complex assembly and enzymatic activity (*Qu et al., 2018*; *Wysocka et al., 2005*). It also recognizes the WIN motif of methyltransferases, such as MLL1, thereby anchoring them to the complex and stabilizing the ASH2L-RbBP5 dimer (*Hsu et al., 2018*). ASH2L further contributes to COMPASS activation by interacting with both RbBP5 and DPY30 and by stabilizing the SET domain, which is essential for efficient substrate recognition and catalysis (*Qu et al., 2018*; *Park et al., 2019*). Our work shows that MgdE interacts with both WDR5 and ASH2L, leading to a decrease in H3K4me3 levels (*Figures 4A–C and 5D*). Site-directed mutagenesis revealed that residues D224 and H247 of MgdE are critical for WDR5 binding, as the double mutant MgdE-D224AH247A fails to interact with WDR5 and shows diminished suppression of H3K4me3 levels (*Figure 5D*).

Moreover, MgdE accumulates in the nucleus (*Figures 1C and 2C*, *Figure 5C*, *Figure 2—figure supplement 1C*), reminiscent of *L. monocytogenes* InlP, which modulates nuclear speckles to inhibit apoptosis (*Pourpre et al., 2022*). This interaction silences the expression of antimicrobial effector genes, thereby facilitating immune evasion. Similar to *L. pneumophila* RomA, which methylates H3K14 to repress innate immune responses (*Schator et al., 2023*; *Rolando et al., 2013*), or Mtb Rv2067c, a structural mimic of host DOT1L that catalyzes non-nucleosomal H3K79 trimethylation to subvert pro-inflammatory signaling (*Singh et al., 2023*), MgdE exemplifies a broader pathogen strategy of hijacking the host histone methylation machinery.

## MgdE-mediated epigenetic silencing suppresses pro-inflammatory responses

Transcriptome analysis showed that the *mgdE*-deleted BCG strain induced hyperactivation of 'Cytokine-cytokine receptor interaction (hsa04060)' and 'JAK-STAT signaling pathway (hsa04630)' (*Figure 6C*), along with significantly elevated production of inflammatory cytokines, including *IL6* and *IL1B*, in THP-1 macrophages compared to the WT BCG strain (*Figures 3C, D and 6E, F*). This profile is consistent with the functional restoration of the COMPASS complex (*Wang and Helin, 2025*; *Yu et al., 2017*). These findings support the hypothesis that MgdE suppresses host inflammatory responses by disrupting COMPASS-mediated H3K4me3 deposition at active promoters. H3K4me3 is a well-established histone mark associated with transcriptional activation, in part through the recruitment of PHD domain-containing reader proteins (*Hyun et al., 2017*). For instance, SET1-dependent deposition of H3K4me3 at NF-κB target promoters facilitates the activation of inflammatory gene expression (*Bhattacharya et al., 2023*). In addition, WDR5 has been implicated in upregulating immunosuppressive cytokines, including *IL6* and *TGFβ* (*Deng et al., 2024*). In transfected HEK293T cells, WT MgdE

significantly reduced global H3K4me3 levels, whereas the catalytically inactive D224AH247A mutant failed to do so (*Figure 5D*), supporting the notion that MgdE-mediated suppression of H3K4me3 depends on its ability to interact with WDR5. Furthermore, during infection, the ΔMgdE strain markedly activated the expression of inflammatory factors compared to the WT BCG strain (*Figure 6E–G*). These results suggest that MgdE attenuates the expression of inflammatory mediators, such as *IL6* and *IL1B* by reducing host H3K4me3 levels during Mycobacterium infection. This mechanism is similar to those observed in other pathogens, such as spirochete-derived factors that suppress inflammation by reducing H3K4me3 levels at the *TNFα* and *IL6* promoters (*Chauhan et al., 2015*).

In summary, our study identified MgdE as a critical mycobacterial nucleomodulin and uncovered a novel paradigm of pathogen-mediated epigenetic regulation through the 'MgdE-COMPASS complex–H3K4me3-cytokine' axis (*Figure 7E*). Mechanistically, MgdE might destabilize the COMPASS assembly via the interaction with ASH2L or WDR5, resulting in reduced H3K4me3 levels at the promoters of pro-inflammatory genes. This epigenetic suppression downregulates cytokine transcription (e.g. *IL6 and IL1B*) and enhances bacterial survival in macrophages and in mice. Our findings reveal the COMPASS complex as a previously unrecognized target of bacterial effectors and offer mechanistic insights into immune evasion and host-pathogen epigenetic interplay.

# Materials and methods

**Key resources table**

| Reagent type (species) or resource | Designation | Source or reference | Identifiers | Additional information |
|---|---|---|---|---|
| Strain, strain background (*Mycobacterium bovis* BCG) | *Mycobacterium bovis* BCG | *Chen et al., 2022* | ATCC: Cat#35734 | |
| Strain, strain background (*Escherichia coli*) | *E. coli* DH5a | *Wang et al., 2025* | ATCC: Cat#25922 | |
| Strain, strain background (Y2HGold Chemically Competent Cell) | Y2HGold | Clontech | Cat#630498 | GAL4-based yeast strain for two-hybrid assays; see 'Materials and methods: Y2H assay' for details. |
| Cell line (*Homo sapiens*) | HEK293T | Cellosaurus | Cat#CRL-3216 RRID:CVCL_0063 | Authenticated and mycoplasma-free; see 'Materials and methods: Cell culture' for details. |
| Cell line (*H. sapiens*) | THP-1 | Cellosaurus | Cat#TIB-202 RRID:CVCL_0006 | Authenticated and mycoplasma-free; see 'Materials and methods: Cell culture' for details. |
| Cell line (*Mus musculus*) | RAW264.7 | Cellosaurus | Cat#TIB-71 RRID:CVCL_C6XG | Authenticated and mycoplasma-free; see 'Materials and methods: Cell culture' for details. |
| Cell line (*M. musculus*) | C57BL/6 | Chang-sheng Bio (Liaoning, China) | RRID:MGI:2159965 | |
| Antibody | Anti-β-actin (Mouse Monoclonal Antibody) | Abbkine | RRID:AB_3740145 | WB 1:10000 |
| Antibody | Anti-Histone H3 (Mouse Monoclonal Antibody) | Abbkine | RRID:AB_3740146 | WB 1:2000 |
| Antibody | Goat anti-rabbit IgG | Abbkine | RRID:AB_2876889 | WB 1:10000 |
| Antibody | Goat anti-mouse IgG | Abbkine | RRID:AB_2737290 | WB 1:1000 |
| Antibody | GAPDH (Mouse Monoclonal Antibody) | Abbkine | RRID:AB_3714704 | WB 1:2000 |
| Antibody | HA tag (Rabbit Polyclonal Antibody) | Proteintech | RRID:AB_11042321 | WB 1:3000 |

*Continued on next page*

*Continued*

| Reagent type (species) or resource | Designation | Source or reference | Identifiers | Additional information |
|---|---|---|---|---|
| Antibody | GFP tag (Rabbit Polyclonal Antibody) | Proteintech | RRID:AB_11182611 | WB 1:5000 |
| Antibody | Flag tag (Mouse Monoclonal Antibody) | Proteintech | RRID:AB_2918475 | WB 1:10000 |
| Antibody | Anti-H3K4me3 (Rabbit Monoclonal Antibody) | Proteintech | RRID:AB_3740151 | WB 1:5000 |
| Antibody | Ag85B | *Zhang et al., 2022* | N/A | WB 1:10000 |
| Antibody | GlpX | *Zhang et al., 2022* | N/A | WB 1:10000 |
| Chemical compound, drug | DMEM | Gibco | Cat#11965092 | Cell culture |
| Chemical compound, drug | RPMI 1640 | Gibco | Cat#C11875500BT | Cell culture |
| Chemical compound, drug | Opti-MEMTM | Gibco | Cat#31985070 | Cell culture |
| Chemical compound, drug | FBS | Gibco | Cat#A5256701 | Cell culture |
| Chemical compound, drug | Sodium pyruvate | Gibco | Cat#11360070 | Cell culture |
| Chemical compound, drug | L-glutamine | Gibco | Cat#A2916801 | Cell culture |
| Chemical compound, drug | HEPES | Gibco | Cat#15630080 | Cell culture |
| Chemical compound, drug | 2-Mercaptoethanol | Gibco | Cat#21985023 | Cell culture |
| Chemical compound, drug | Penicillin-streptomycin antibiotics | Gibco | Cat#15140122 | Cell culture |
| Chemical compound, drug | PMA | Sigma-Aldrich | Cat#P8139 | Cell culture |
| Chemical compound, drug | DO Supplement -Leu/-Trp | Coolaber | Cat#PM2220 | Cell culture |
| Chemical compound, drug | DO Supplement -Ade-His-Leu-Trp | Coolaber | Cat#PM2110 | Y2H strains culture |
| Chemical compound, drug | Aureobasidin A | Coolaber | Cat#CA2332G | Y2H strains culture |
| Chemical compound, drug | OADC | MilliporeSigma | Cat#M0678 | Y2H strains culture |
| Chemical compound, drug | NP-40 | Beyotime | Cat#P0038 | Immunoblotting assay |
| Chemical compound, drug | RIPA buffer | Beyotime | Cat#P0038 | Immunoblotting assay |
| Chemical compound, drug | Protease inhibitor cocktail | Boster | Cat#AR1182 | Immunoblotting assay |
| Chemical compound, drug | Triton-X-100 | Beyotime | Cat#P0096 | Immunoblotting assay |
| Commercial assay or kit | Nuclear and Cytoplasmic Protein Extraction kit | Beyotime | Cat#P0027 | Immunoblotting assay |
| Chemical compound, drug | DAPI | Beyotime | Cat#P0126 | confocal microscopy assay |
| Commercial assay or kit | Clarity Western ECL Substrate | BIO-RAD | Cat#1705060 | confocal microscopy assay |
| Chemical compound, drug | Protein A/G magnetic beads | MedChemExpress | Cat#HY-K0202 | Co-IP assay |
| Chemical compound, drug | Protease inhibitor cocktail | Boster | Cat#AR1182 | Co-IP assay |
| Chemical compound, drug | HieffTrans Liposomal Transfection Reagent | YEASEN | Cat#40802ES03 | Cell fractionation |
| Commercial assay or kit | EndoFree Plasmid Mini Kit | Aidlab | Cat#PL0401 | Cell fractionation |
| Commercial assay or kit | TRIpure Reagent | Aidlab | Cat#RN0101 | qRT-PCR assay |
| Commercial assay or kit | EASYspin RNA Mini Kit | Aidlab | Cat#RN0702 | qRT-PCR assay |
| Commercial assay or kit | cDNA Reverse Transcription Kit | Vazyme | Cat#R333-01 | qRT-PCR assay |
| Commercial assay or kit | 2× ChamQ Universal SYBR qPCR Master Mix | Vazyme | Cat#Q711-03 | qRT-PCR assay |

*Continued*

| Reagent type (species) or resource | Designation | Source or reference | Identifiers | Additional information |
|---|---|---|---|---|
| Software, algorithm | AlphaFold v2.2.0 | SciCrunch Registry | RRID:SCR_023662 | |
| Software, algorithm | GraphPad Prism 8 | SciCrunch Registry | RRID:SCR_002798 | |
| Software, algorithm | UCSF Chimera | SciCrunch Registry | RRID:SCR_004097 | |
| Software, algorithm | MEGA 12.0 | SciCrunch Registry | RRID:SCR_000667 | |
| Software, algorithm | Cytoscape 3.10.3 | SciCrunch Registry | RRID:SCR_003032 | |
| Software, algorithm | CaseViewer v2.0 | SciCrunch Registry | RRID:SCR_017654 | |
| Software, algorithm | jvenn/Venny 2.1.0 | SciCrunch Registry | RRID:SCR_016343 | |
| Software, algorithm | cNLS Mapper | N/A | N/A | |
| Software, algorithm | iTOL | SciCrunch Registry | RRID:SCR_018174 | |
| Software, algorithm | ESPript | SciCrunch Registry | RRID:SCR_006587 | |
| Software, algorithm | SignalP 5.0 | SciCrunch Registry | RRID:SCR_015644 | |
| Software, algorithm | SecretomeP 2.0 | SciCrunch Registry | RRID:SCR_026505 | |
| Software, algorithm | STRING 12.0 | SciCrunch Registry | RRID:SCR_005223 | |

## Bacterial strains and culture conditions

The bacterial strains used in this study are detailed in **Supplementary file 6**. *Escherichia coli* DH5α was cultured in Luria-Bertani medium under standard conditions. *Mycobacterium bovis BCG* strains were grown in Middlebrook 7H9 broth (BD Biosciences), containing 1x OADC (oleic acid, albumin, dextrose, catalase), 0.05% Tween-80, and 2% glycerol. Antibiotics were added as required at the following final concentrations: 50 µg/mL hygromycin (for mycobacteria) and 50 µg/mL kanamycin (for both mycobacteria and *E. coli*). The Y2H Gold yeast strain (Takara Bio) was maintained in YPDA (yeast extract, peptone, dextrose, and adenine sulfate) medium. Transformants were selected on SD/-Trp medium, which lacks tryptophan, to select for plasmid uptake. Protein-protein interactions were assessed in SD/-Trp/-Leu (lacking tryptophan and leucine) and SD/-Trp/-Leu/-Ade/-His (lacking tryptophan, leucine, adenine, and histidine) media, with growth on higher stringency media indicating positive interactions.

## Cell culture

THP-1 cells were cultured in Roswell Park Memorial Institute 1640 (RPMI 1640) medium supplemented with 10% (v/v) heat-inactivated fetal bovine serum (FBS), 1 mM sodium pyruvate, 2 mM L-glutamine, 10 mM HEPES buffer (pH 7.2–7.5), and 50 µM 2-mercaptoethanol. For differentiation into macrophages, THP-1 cells were treated with 200 nM phorbol 12-myristate 13-acetate (PMA) for 48 hr to allow complete differentiation prior to their use in experiments.

HEK293T and RAW264.7 monocytes were cultured in Dulbecco's modified Eagle's medium (DMEM) supplemented with 10% FBS and 50 µg/mL penicillin-streptomycin. All cells were maintained at 37 °C in a humidified atmosphere with 5% $CO_2$.

These cell lines were purchased from and authenticated by American Type Culture Collection (ATCC) (Key resources table), and were tested negative for mycoplasma contamination.

## Screening of conserved nucleomodulins in mycobacteria

To systematically identify evolutionarily conserved nucleomodulins in pathogenic mycobacteria, a comparative pan-genome analysis was performed across four phylogenetically representative mycobacterium strains: *Mycobacterium tuberculosis* H37Rv (NCBI RefSeq assembly GCF_000195955.2), *Mycobacterium tuberculosis* H37Ra (NCBI RefSeq assembly GCF_001938725.1), *Mycobacterium bovis* BCG (NCBI RefSeq assembly GCF_000009445.1), *Mycobacterium marinum* (NCBI RefSeq assembly GCF_000018345.1), and *Mycobacterium avium* subsp (NCBI RefSeq assembly GCF_009741445.1). Complete genome sequences and annotations were obtained from NCBI RefSeq with strict inclusion criteria requiring 'Complete Genome' status and fewer than 50 contigs to ensure the assembly

integrity. Plasmid sequences, as well as low-complexity and repetitive regions, were filtered using the NCBI Genome Data Viewer toolkit. Genome quality control was rigorously conducted using CheckM with a *Mycobacteriaceae*-specific marker set, confirming that all the selected genomes exhibited >99% completeness and <1% contamination. Ortholog clustering of protein-coding sequences was performed using OrthoFinder with DIAMOND for sequence alignment. Orthogroups were inferred using the Markov Cluster Algorithm (MCL) with an inflation parameter of 1.5 to balance sensitivity and specificity. Core orthologs were defined as groups present in all four strains, with additional filtering to retain only single-copy genes in ≥90% of the strains, thereby minimizing paralog-related artifacts.

To prioritize candidate nucleomodulins, a sequential screening pipeline was applied, beginning with the prediction of classical secretory proteins using SignalP v5.0 (D-score≥0.5), followed by the identification of non-classically secreted proteins via SecretomeP v2.0 (NN score ≥0.9), and culminating in NLS detection using cNLS Mapper (NLS score ≥2.0), which predicts both monopartite and bipartite NLS. This integrated strategy enabled the systematic identification of conserved candidates that exhibit both secretory potential and nuclear targeting capacity.

## Immunoblot analysis of MgdE secretion

The procedure for immunoblot analysis of MgdE secretion was adapted from *Zhang et al., 2022* with minor modifications. *M. bovis* BCG strains expressing C-terminally Flag-tagged MgdE were cultured in Middlebrook 7H9 broth supplemented with 0.5% glycerol, 0.02% Tyloxapol, and 50 μg/mL kanamycin. Cultures were incubated at 37 °C with gentle agitation (80 rpm) until reaching an $OD_{600}$ of ~0.6. Cells were harvested by centrifugation at 4000×*g* for 15 min at 4 °C. The culture supernatants were filtered through 0.22 μm PES membranes (Millipore) to remove residual bacteria and subsequently concentrated ~100 fold using Amicon Ultra-15 centrifugal filters with a 10 kDa molecular weight cutoff (Millipore). The resulting culture filtrates were used for detection of secreted MgdE. Bacterial pellets were washed twice with PBS and resuspended in lysis buffer containing 50 mM Tris-HCl (pH 7.5), 150 mM NaCl, 1 mM PMSF, and 1x protease inhibitor cocktail. Cells were lysed by sonication on ice, and the lysates were cleared by centrifugation at 10,000×*g* for 15 min at 4 °C to obtain whole-cell lysates. Whole-cell lysates and concentrated culture filtrates were resolved on 10% SDS-PAGE gels and transferred to PVDF membranes (Millipore). Membranes were blocked with 5% skimmed milk powder in TBST (Tris-buffered saline containing 0.1% Tween-20) for 1 hr at room temperature and then incubated overnight at 4 °C with a monoclonal anti-Flag antibody. After washing, membranes were incubated with HRP-conjugated secondary antibodies for 1 hr at room temperature. Immunoreactive bands were visualized using enhanced chemiluminescence (ECL) substrate and imaged with a ChemiDoc MP imaging system (Bio-Rad).

## Phylogenetic and sequence analysis

To investigate the evolutionary conservation and sequence features of MgdE homologs, phylogenetic reconstruction was performed using MEGA 12.0 with a neighbor-joining algorithm. The parameters included pairwise deletion for gap treatment, a Poisson substitution model (selected for its suitability in analyzing closely related bacterial sequences), and 1000 bootstrap replicates to assess node support. Nodes with ≥50% bootstrap values were retained, and no additional pruning was performed.

The analysis incorporated MgdE homologs from 12 Mycobacterium strains spanning clinical, environmental, and attenuated lineages. These included *M. tuberculosis* (Mtu), *M. bovis* BCG (Mbb), *M. tuberculosis* H37Ra (Mra), *M. africanum* (Maf), *M. tuberculosis* CDC1551 (Mtc), *M. tuberculosis* KZN 1435 (Mtb), *M. shinjukuense* (Mshj), *M. marinum* M (Mmi), *M. liflandii* (Mli), *M. intracellulare subsp* (Mid), *M. avium* (Mav), *M. mantenii* (Mman). Genomic sequences were retrieved from the NCBI RefSeq database, filtered for completeness, and aligned using Clustal Omega v1.2.4 with default parameters (gap opening penalty = 10, gap extension = 0.2, and iterative refinement enabled). Sequence conservation was visualized using ESPript 3.0, highlighting residues with ≥90% identity across all homologs in red.

## Computational prediction of sequence and structural features

AlphaFold v2.2.0 was used to predict the interaction between MgdE and COMPASS complex subunits, following the methodology established in a previous study (*Gómez Borrego and Torrent Burgas, 2024*). The specific parameters used for the prediction were as follows: model_preset = multimer;

db_preset = full_dbs. A FASTA file containing the sequences of both MgdE and COMPASS complex subunit proteins was input into the AlphaFold software. The resulting models were evaluated and ranked based on the AF-score, which is a linear combination of the interface score (ipTM) and the predicted TM-score (pTM). The top-ranked model (rank 1) was selected for further analysis.

## Cell transfection and confocal microscopy

HEK293T cells were seeded at 70% confluency in poly-L-lysine-coated 35 mm glass-bottom dishes and maintained in DMEM supplemented with 10% FBS at 37 °C under 5% $CO_2$. Transfection was performed using the Hieff Trans Liposomal Transfection Reagent (Yeasen) according to the manufacturer's protocol. Briefly, 0.5 µg of DNA and 1.5 µL of transfection reagent were mixed with 50 µL of Opti-MEM and incubated for 20 min at room temperature. The mixture was added dropwise to each chamber of 35 mm six-chamber glass-bottom dishes containing cells at 40-60% confluency.

At 4–48 hr post-transfection, the cells were washed with PBS, fixed in 4% paraformaldehyde for 15 min, and permeabilized with 0.1% Triton X-100 for 10 min. Nuclei were counterstained with 1 µg/mL DAPI for 10 min. Confocal imaging was performed using an Olympus FV1000 Confocal Laser Scanning Microscope equipped with a 100x/1.40 NA oil immersion objective. EGFP and DAPI signals were detected with 488 nm and 405 nm lasers, respectively. Spectral unmixing was applied using the Olympus FV10-ASW software (v4.2) to minimize signal cross-talk, and Z-stacks were acquired in sequential mode by applying a step size of 0.5 µm across all samples to ensure comparability.

## Cell fractionation and immunoblotting

During infection, THP-1 cells were differentiated with 200 ng/mL PMA for 48 hr, followed by infection with BCG/MgdE-Flag, BCG/MgdE$^{\Delta NLS1}$-Flag, BCG/MgdE$^{\Delta NLS2}$-Flag, or BCG/MgdE$^{\Delta NLS1-2}$-Flag strains at a multiplicity of infection (MOI) of 50 for 24 hr. Cell fractionation was performed with minor modifications based on a previously described method (Wang J et al., 2017). Briefly, cells were lysed in a hypotonic buffer containing 10 mM HEPES (pH 7.9), 1.5 mM $MgCl_2$, 10 mM KCl, 0.34 M sucrose, 10% glycerol, and 0.02% Tyloxapol, supplemented with 1% protease inhibitor cocktail, and incubated for 10 min at 4 °C. The lysates were then centrifuged at 1300×$g$ for 10 min. The pellet from uninfected THP-1 cells was collected as the nuclear fraction. For infected THP-1 cells, the pellet was resuspended in PBS containing 0.1% SDS and centrifuged at 2500×$g$ for 15 min at 4 °C. The resulting supernatant was collected as the nuclear fraction.

During transfection, HEK293T cells were cultured in DMEM supplemented with 10% FBS at 37 °C in a 5% $CO_2$ incubator. Transfection was performed as described in the 'Cell transfection and confocal microscopy' section. At 36 hr post-transfection, nuclear and cytoplasmic protein fractionation was conducted using the Nuclear and Cytoplasmic Protein Extraction Kit (Beyotime) with minor modifications. Cells were washed with PBS, detached using a cell scraper, and pelleted by centrifugation at 2000×$g$ for 5 min. After removing the supernatant, 150 µL of Cytoplasmic Extraction Reagent A (with PMSF) was added to ~20 µL of packed cells, vortexed for 5 s, and incubated on ice for 15 min. Then, 10 µL of Reagent B was added, followed by vortexing and ice incubation. The mixture was centrifuged at 12,000×$g$ for 5 min at 4 °C, and the supernatant was collected as the cytoplasmic fraction. The pellet was resuspended in 50 µL of Nuclear Extraction Reagent (with PMSF), vortexed intermittently over 30 min on ice, and centrifuged at 12,000×$g$ for 10 min at 4°C. The resulting supernatant was collected as the nuclear protein fraction.

Protein concentrations were quantified using the BCA assay (Thermo Fisher Scientific). Equal amounts of protein (20-40 µg) were resolved by sodium dodecylsulfate polyacrylamide gel electrophoresis (SDS-PAGE) and transferred onto PVDF membranes. The membranes were blocked for 1 hr at room temperature with 5% non-fat milk in TBST and incubated overnight at 4 °C with primary antibodies. After three washes with TBST, the membranes were incubated with HRP-conjugated secondary antibodies at room temperature for 1 hr. Following three additional TBST washes, protein signals were detected using an ECL substrate and imaged on a ChemiDoc MP system (Bio-Rad). The antibody information is listed in the Key resources table.

## Co-IP assays

The Co-IP assays were performed to analyze protein-protein interactions. HEK293T cells were washed twice with ice-cold PBS and lysed in lysis buffer (50 mM Tris-HCl, pH 7.4, 150 mM NaCl, 1% NP-40)

supplemented with 1.5% protease inhibitor cocktail (Boster) for 30 min on ice. Crude lysates were centrifuged at 12,000×$g$ for 15 min at 4 °C to remove the debris. For immunoprecipitation, 1 mg of total protein was incubated with 5 μg of primary antibody overnight at 4 °C with gentle agitation. Protein A/G magnetic beads (MedChemExpress) were added to the antibody-protein complexes and incubated for 6 hr at 4 °C. The beads were washed three times with PBS-T buffer (PBS containing 0.5% Tween-20) to remove nonspecific interactions. Proteins were eluted by boiling the beads in 1x SDS sample buffer at 95 °C for 5 min. The eluted proteins were resolved by SDS-PAGE and analyzed by immunoblotting. Antibody information is listed in the Key resources table.

## Y2H assay

The Yeast Two-Hybrid System (Clontech) was used to assay protein-protein interactions between MgdE (WT and mutants) and subunits of the COMPASS complex (ASH2L, WDR5, RbBP5, and DPY30). Full-length cDNAs encoding WT MgdE and its mutants (see *Supplementary file 5*) were cloned into the bait vector pGBKT7, whereas genes encoding the COMPASS subunits were cloned into the prey vector pGADT7. The yeast strain Y2H Gold was co-transformed with bait and prey plasmids using the lithium acetate/polyethylene glycol method, and the transformed colonies were initially selected on SD agar plates lacking leucine and tryptophan (-Leu/-Trp) to confirm plasmid retention. For interaction screening, colonies were replica-plated onto selective SD agar plates lacking leucine, tryptophan, adenine, and histidine (-Leu/-Trp/-Ade/-His) and supplemented with 200 ng/μL aureobasidin A (AbA, Sigma-Aldrich). The plates were incubated at 30 °C, and colony growth was monitored daily, with final assessments performed after 5 days of incubation. Interaction strength was qualitatively assessed by comparing colony density and growth rates with those of the controls. The positive controls included the co-transformation of pGBKT7-p53 with pGADT7-T, which produced robust growth on -Leu/-Trp/-Ade/-His plates. Negative controls included co-transformation of pGBKT7-Lam (human lamin C, a non-interacting bait) with pGADT7-T, as well as individual transformations with empty pGBKT7 or pGADT7 vectors to assess the background growth. The yeast strains used in this study are listed in *Supplementary file 6*.

## RNA-seq analysis

THP-1 cells were seeded at a density of 1×10⁶ cells/well in 6-well plates and differentiated into macrophages by treatment with 200 nM PMA for 48 hr. Following differentiation, the cells were washed with PBS and infected with *M. bovis* BCG WT or ΔMgdE strains at a MOI of 10:1. Following 24 hr of infection, total RNA was extracted using TRIzol reagent (Thermo Fisher Scientific) according to the manufacturer's instructions. High-quality reads were then mapped to the human reference genome (GRCh38) using HISAT2 (v2.2.1). FeatureCounts (v2.0.3) was used to quantify transcript abundance, and differential gene expression analysis was conducted using DESeq2 (v1.38.3) in R. Genes with an adjusted $p$-value<0.05 and an absolute $\log_2$-fold change ≥1 was considered significantly differentially expressed. RNA-seq data were visualized using GraphPad Prism 8.0, and volcano plots, heatmaps, and other visualizations were generated to summarize the results.

## qRT-PCR assay

THP-1 cells were infected with the *M. bovis* BCG strain and harvested 24 hr post-infection for RNA isolation. Total RNA was extracted using 1 mL of TRIzol reagent (Aidlab) according to the manufacturer's instructions. To remove genomic DNA contamination, the RNA samples were treated with DNase I (Thermo Fisher Scientific) for 30 min at 37 °C, followed by heat inactivation at 65°C for 10 min. RNA concentration and purity were assessed using a NanoDrop 2000 spectrophotometer (Thermo Fisher Scientific). For cDNA synthesis, 1 μg of total RNA was reverse transcribed using the HiScript III cDNA Reverse Transcription Kit (Vazyme) with random hexamers, according to the manufacturer's protocol. qRT-PCR was performed using ChamQ SYBR Green RT-PCR Master Mix (Vazyme). The relative mRNA expression of different genes was calculated by comparing their cycle threshold (Ct) values to that of a control gene *HPRT* using the 2$^{-\Delta\Delta Ct}$ method. Real-time qRT-PCR experiments were performed in triplicate, with at least three independent biological replicates and two technical replicates per condition. Primer sequences used for qRT-PCR are listed in *Supplementary file 4*.

## CFU assay

For bacterial preparation, mid-log phase cultures ($OD_{600}$=0.8) of *M. bovis* BCG were washed with PBS and resuspended in fresh medium. THP-1 cells were seeded at a density of $5\times10^5$ cells/well in 24-well plates and subsequently infected at a MOI of 10:1. Following 4 hr of infection, extracellular bacteria were eliminated by treatment with penicillin-streptomycin (100 μg/mL) for 2 hr. The cells were then washed three times with PBS and maintained in an antibiotic-free medium until harvest. For bacterial recovery, the macrophages were lysed with 0.25% SDS for 10 min at room temperature, followed by serial dilution of the lysates in PBS. Aliquots (100 μL) of each dilution were plated on Middlebrook 7H10 agar supplemented with 10% OADC and kanamycin (50 μg/mL). Plates were incubated at 37 °C for 14 days, and CFUs were calculated by applying dilution factors to the colony counts. *M. bovis* BCG strains used in this study are listed in *Supplementary file 6*.

## Animal experiments

Specific pathogen-free (SPF) female C57BL/6 mice (6–8 weeks old) were obtained from Chang-sheng Bio (Liaoning, China). All animals were housed under SPF conditions under controlled temperature, humidity, and a 12 hr light/dark cycle, with free access to food and water.

Mice were infected as described previously with slight modifications (*Chen et al., 2022*). Mid-log phase *M. bovis* BCG cultures were washed twice in PBS containing 0.05% Tween 80 and sonicated to disrupt clumps. Female C57BL/6 mice (6–8 weeks old, 16–18 g) were anesthetized using isoflurane and intranasally administered $1\times10^7$ CFUs of *M. bovis* BCG in 40 μL of PBS. At 2 days post-infection, the bacterial load in the lungs of six mice was assessed to confirm the inoculation dose. The lungs were homogenized in 1 mL of PBS using a tissue homogenizer, and serial dilutions of the homogenates were plated onto Middlebrook 7H10 agar with 10% OADC. The plates were then incubated at 37 °C with 5% $CO_2$ for 14 days prior to colony counting.

## Histological analysis

Lung tissues from *M. bovis* BCG-infected mice were fixed in 4% phosphate-buffered formalin at room temperature for 24 hr and embedded in paraffin wax. Paraffin-embedded tissue samples were sectioned into 2–3 μm thick slices using a microtome. The sections were deparaffinized, rehydrated with graded ethanol, and stained with hematoxylin and eosin (H&E). Stained slides were visualized using an Olympus BX53 light microscope, and CaseViewer version 2.0 (3DHISTECH) was used to view and annotate the scanned slides.

## Statistical analyses

Data are presented as mean ± standard deviation (SD). Statistical significance was assessed using unpaired *t*-tests or two-way ANOVA. Statistical significance was set at $p<0.05$. Significance levels are denoted in the figures as follows: $p>0.05$ (ns, not significant), $p<0.05$ (*), $p<0.01$ (**), and $p<0.001$ (***). All statistical analyses were performed using GraphPad Prism (version 8.0). Data were collected from at least three independent biological replicates, each with two or more technical replicates.

# Acknowledgements

This work was supported by the National Key Research and Development Program of China (#2021YFD1800403), Fundamental Research Funds for the Central Universities (#2662023DKQD001), Talent Start-up Funds of Huazhong Agricultural University (#11042310008), National Natural Science Foundation of China (#32473123), China Agriculture Research System of MOF and MARA (#CARS-37).

## Additional information

### Funding

| Funder | Grant reference number | Author |
| --- | --- | --- |
| National Key Research and Development Program of China | 2021YFD1800403 | Yingyu Chen |
| Fundamental Research Funds for the Central Universities | 2662023DKQD001 | Lei Zhang |
| Talent Start-up Funds of Huazhong Agricultural University | 11042310008 | Lei Zhang |
| National Natural Science Foundation of China | 32473123 | Aizhen Guo |
| China Agriculture Research System of MOF and MARA | CARS-37 | Aizhen Guo |

The funders had no role in study design, data collection and interpretation, or the decision to submit the work for publication.

### Author contributions

Liu Chen, Conceptualization, Data curation, Formal analysis, Validation, Investigation, Visualization, Methodology, Writing – original draft, Writing – review and editing; Baojie Duan, Validation, Investigation; Pingping Chen, Investigation; Qiang Jiang, Data curation, Validation, Investigation, Visualization; Yifan Wang, Data curation, Formal analysis, Investigation; Lu Lu, Data curation, Investigation; Yingyu Chen, Resources, Supervision, Funding acquisition, Visualization, Project administration; Changmin Hu, Supervision, Visualization, Project administration; Lei Zhang, Aizhen Guo, Conceptualization, Resources, Supervision, Funding acquisition, Visualization, Writing – original draft, Project administration, Writing – review and editing

### Author ORCIDs

Liu Chen ⓘ https://orcid.org/0009-0003-3722-6994
Yingyu Chen ⓘ https://orcid.org/0000-0002-1200-5314
Lei Zhang ⓘ https://orcid.org/0000-0002-8566-6068

### Ethics

All animal experiments were conducted in accordance with protocols approved by the Animal Ethics Committee of Huazhong Agricultural University and carried out under the guidelines of the Institutional Animal Care and Use Committee, in compliance with license number SYXK (Hubei) 2020-0084.

Reviewer #1 (Public review): https://doi.org/10.7554/eLife.107677.4.sa1
Reviewer #2 (Public review): https://doi.org/10.7554/eLife.107677.4.sa2
Reviewer #3 (Public review): https://doi.org/10.7554/eLife.107677.4.sa3
Author response https://doi.org/10.7554/eLife.107677.4.sa4

## Additional files

### Supplementary files

MDAR checklist

Supplementary file 1. Five Mycobacterium species secreted proteins.

Supplementary file 2. Five Mycobacterium species with conserved nucleomodulins.

Supplementary file 3. 56 candidate protein nuclear phenotypes.

Supplementary file 4. Primers used in this study.

Supplementary file 5. Plasmids used in this study.

Supplementary file 6. Bacterial strains used in this study.

## Data availability

RNA sequence data were deposited in GEO (accession number GSE312039).

The following dataset was generated:

| Author(s) | Year | Dataset title | Dataset URL | Database and Identifier |
|---|---|---|---|---|
| Liu C, Duan B, Jiang Q, Wang Y, Chen Y, Hu C, Zhang L, Guo A | 2025 | Differential transcriptional effects of mycobacterial nucleomodulins MgdE and MmpE in THP-1 macrophages | https://www.ncbi.nlm.nih.gov/geo/query/acc.cgi?acc=GSE312039 | NCBI Gene Expression Omnibus, GSE312039 |

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
