## [Editor Report · eLife Assessment]

This **valuable** study provides **convincing** evidence that MgdE, a conserved mycobacterial nucleomodulin, downregulates inflammatory gene transcription by interacting with the histone methyltransferase COMPASS complex and altering histone H3 lysine methylation. This work will interest microbiologists as well as cell and cancer biologists.

---

## [Referee Report · Reviewer #1 (Public review)]

Summary:

This fundamental study identifies a new mechanism that involves a mycobacterial nucleomodulin manipulation of the host histone methyltransferase COMPASS complex to promote infection. Although other intracellular pathogens are known to manipulate histone methylation, this is the first report demonstrating specific targeting the COMPASS complex by a pathogen. The rigorous experimental design using of state-of-the art bioinformatic analysis, protein modeling, molecular and cellular interaction and functional approaches, culminating with in vivo infection modeling provide convincing, unequivocal evidence that supports the authors claims. This work will be of particular interest to cellular microbiologist working on microbial virulence mechanisms and effectors, specifically nucleomodulins, and cell/cancer biologists that examine COMPASS dysfunction in cancer biology.

Strengths:

(1) The strengths of this study include the rigorous and comprehensive experimental design that involved numerous state-of-the-art approaches to identify potential nucleomodulins, define molecular nucleomodulin-host interactions, cellular nucleomodulin localization, intracellular survival, and inflammatory gene transcriptional responses, and confirmation of the inflammatory and infection phenotype in a small animal model.

(2) The use of bioinformatic, cellular and in vivo modeling that are consistent and support the overall conclusions is a strengthen of the study. In addition, the rigorous experimental design and data analysis including the supplemental data provided, further strengthens the evidence supporting the conclusions.

Comments on revisions:

The authors have previously addressed the weaknesses that were identified by this reviewer by providing rational explanation and specific references that support the findings and conclusions.

---

## [Referee Report · Reviewer #2 (Public review)]

Summary:

The manuscript by Chen et al addresses an important aspect of pathogenesis for mycobacterial pathogens, seeking to understand how bacterial effector proteins disrupt the host immune response. To address this question the authors sought to identify bacterial effectors from *M. tuberculosis* (Mtb) that localize to the host nucleus and disrupt host gene expression as a means of impairing host immune function. Their revised manuscript has strengthened their observations by performing additional experiments with BCG strains expressing tagged MgdE.

Strengths:

The researchers conducted a rigorous bioinformatic analysis to identify secreted effectors containing mammalian nuclear localization signal (NLS) sequences, which formed the basis of quantitative microscopy analysis to identify bacterial proteins that had nuclear targeting within human cells. The study used two complementary methods to detect protein-protein interaction: yeast two-hybrid assays and reciprocal immunoprecipitation (IP). The combined use of these techniques provides strong evidence of interactions between MgdE and SET1 components and suggests the interactions are in fact direct. The authors also carried out rigorous analysis of changes in gene expression in macrophages infected with MgdE mutant BCG. They found strong and consistent effects on key cytokines such as IL6 and CSF1/2, suggesting that nuclear-localized MgdE does in fact alter gene expression during infection of macrophages. The revised manuscript contains additional biochemical analyses of BCG strains expressing tagged MgdE that further supports their microscopy findings.

---

## [Referee Report · Reviewer #3 (Public review)]

In this study, Chen L et al. systematically analyzed the mycobacterial nucleomodulins and identified MgdE as a key nucleomodulin in pathogenesis. They found that MgdE enters into host cell nucleus through two nuclear localization signals, KRIR108-111 and RLRRPR300-305, and then interacts with COMPASS complex subunits ASH2L and WDR5 to suppress H3K4 methylation-mediated transcription of pro-inflammatory cytokines, thereby promoting mycobacterial survival.

Comments on revisions:

The authors have previously adequately addressed previous concerns through additional experimentation. The revised data robustly support the main conclusions, demonstrating that MgdE engages the host COMPASS complex to suppress H3K4 methylation, thereby repressing pro-inflammatory gene expression and promoting mycobacterial survival. This work represents a significant conceptual advance.

---

## [Author Response]

The following is the authors’ response to the previous reviews

**Reviewer #2 (Recommendations for the authors):**
Major:Over-interpretation of data. There are a few instances of this:The authors claim "Our work shows that MgdE interacts with both WDR5 and ASH2L and inhibits the methyltransferase activity of the COMPASS complex" (Line 318). However, they provide no biochemical analysis of methyltransferase activity to support this claim. While they cite Figure 4A-C and Figure 5, these data simply show (slightly) decreased cellular levels of H3K4Me. There are multiple ways H3K4Me could decrease including blocking recruitment of COMPASS to promoters or the enzymatic activity of MgdE itself.The data itself related to H3K4Me changes (Figure 5D) is difficult to interpret in light of the controls they now provide. Examining the blot itself there seems to be a massive increase in H3K4Me in control cells expressing GFP that is not reflected in the quantification that shows only a ~2x increase in GFP-expressing cells. In addition， there is very little decrease in H3K4Me in the MgdE-expressing cells relative to controls or site-mutant (no change apparent visually and ~10% change per their quantification). However, the authors interpret this as," revealed that cells expressing WT MgdE exhibited lower levels of H3K4me3". In both these cases I would recommend the authors consider modifying their interpretation of the data.

We thank the reviewer for the comment.

(1) We have now revised this interpretation in the manuscript as follows:

Lines 311-312: “Our work shows that MgdE interacts with both WDR5 and ASH2L, leading to a decrease in H3K4me3 levels.”

(2) Figure 5D presents the results of three independent biological replicates. The bar graph shows the average signal intensity of H3K4me3 normalized to the corresponding loading controls. Accordingly, we have revised the analysis and description of the experimental results.

Lines 214-217: “Immunoblot analysis of nuclear extracts showed that cells expressing WT MgdE had ~25% lower H3K4me3 levels than EGFP-expressing cells and ~40% lower levels than those expressing the D244A/H47A mutant (Figure 5D).”

MinorWhat is "CK"? Please clarify (Figure 2F).

We thank the reviewer for the comment. In this context, "CK" refers to the uninfected control group, which serves as the negative control in the experiment. We have revised the label in Figure 2F.

How many times was the BCG mouse experiment performed? This should be indicated in the figure legend? (Figure 7A).

We thank the reviewer for the comment. The BCG mouse experiment was performed once, and we have added this information to the figure legend of Figure 7A.

It is unclear why the secreted protein (after signal peptide removal) migrates at the same size as the full-length protein (Figure S2).

We thank the reviewer for the comment. The precursors of secreted proteins after translation in the cytoplasm will be translated into the periplasm immediately. Therefore, MgdE or Ag85B obtained from the whole-cell lysate (Figure S2A) mostly have had the signal peptides removed. This is also validated in the case of Rv0455c secretion by Mtb (Zhang et al., Nature Communications, 2022). This explains why MgdE (or Ag85B) proteins from whole-cell lysates or from supernatants show same size in SDS-PAGE gels.

It is still unclear why the transcripts with very little fold-change in expression (in grey) have the most significant p-values for being different (Figure 6).

We thank the reviewer for the comment. The p-value calculation takes into account not only the magnitude of expression change but also the consistency of expression levels within each group and the number of biological replicates. When the variation among replicates is minimal, even a small difference in group means can result in a statistically significant *p*-value. In our RNA-seq analysis, we used DESeq2 with three biological replicates per group. DESeq2 employs a model based on the negative binomial distribution and accounts for multiple factors, including the mean expression level, within-group variance (dispersion), sample size, and normalization accuracy. As a result, it is common to observe that genes with small variability and strong consistency between replicates may show significant *p*-values even with modest fold changes. Conversely, genes with larger fold changes but greater variability might not reach statistical significance.

Reference

Zhang L, Kent JE, Whitaker M, Young DC, Herrmann D, Aleshin AE, Ko YH, Cingolani G, Saad JS, Moody DB, Marassi FM, Ehrt S, Niederweis M (2022) A periplasmic cinched protein is required for siderophore secretion and virulence of *Mycobacterium tuberculosis* Nat Commun 13(1):2255.